# Evaluation of Isoprene Nitrate Chemistry in Detailed Chemical Mechanisms

Alfred W. Mayhew[1], Ben H. Lee[2], Joel A. Thornton[2], Thomas J. Bannan[3], James Brean,[4] James R. Hopkins[1,5], James D. Lee[1,5], Beth S. Nelson[1], Carl Percival[3], Andrew R. Rickard[1,5], Marvin D. Shaw[1,5], Peter M. Edwards[1], Jaqueline F. Hamilton[1]

[1]Wolfson Atmospheric Chemistry Laboratories, Department of Chemistry, University of York, Heslington, York, UK
[2]Department of Atmospheric Sciences, University of Washington Seattle, Washington 98195, USA
[3]School of Earth and Environmental Sciences, University of Manchester, Manchester, UK
[4]School of Geography, Earth and Environmental Sciences, University of Birmingham, Birmingham, U.K.
[5]National Centre for Atmospheric Science, University of York, York, UK

*Correspondence to*: Jaqueline F. Hamilton (jacqui.hamilton@york.ac.uk)

**Abstract.** Isoprene nitrates are important chemical species in the atmosphere which contribute to the chemical cycles that form ozone and secondary organic aerosol (SOA) with implications for climate and air quality. Accurate chemical mechanisms are important for the prediction of the atmospheric chemistry of species such as isoprene nitrates in chemical models. In recent years, studies into the chemistry of isoprene nitrates have resulted in the development of a range of mechanisms available for use in the simulation of atmospheric isoprene oxidation. This work uses a 0-D chemical box-model to assess the ability of three chemically detailed mechanisms to predict the observed diurnal profiles of four groups of isoprene-derived nitrates in the summertime in the Chinese Megacity of Beijing. An analysis of modelled $C_5H_9NO_5$ isomers, including isoprene hydroperoxy nitrate (IPN) species, highlights the significant contribution of non-IPN species to the $C_5H_9NO_5$ measurement, including the potentially large contribution of nitrooxy hydroxyepoxide (INHE). The changing isomer distribution of isoprene hydroxy nitrates (IHN) derived from OH-initiated and $NO_3$-initiated chemistry is discussed, as is the importance of up-to-date alkoxy radical chemistry for the accurate prediction of isoprene carbonyl nitrate (ICN) formation. All mechanisms under-predicted $C_4H_7NO_5$ as predominately formed from the major isoprene oxidation products, methyl vinyl ketone (MVK) and methacrolein (MACR). This work explores the current capability of existing chemical mechanisms to accurately represent isoprene nitrate chemistry in urban areas significantly impacted by anthropogenic and biogenic chemical interactions. It suggests considerations to be taken when investigating isoprene nitrates in ambient scenarios, investigates the potential impact of varying isomer distributions on iodide chemical ionisation mass spectrometry ($I^-$-CIMS) calibrations, and makes some proposals for the future development of isoprene mechanisms.

## 1 Introduction

Isoprene (2-methyl-1,3-butadiene) is the most emitted non-methane volatile organic compound (NMVOC) globally, and accounts for around 70% of global biogenic volatile organic compound (BVOC) emissions.(Guenther et al., 1995; Guenther

et al., 2006; Guenther et al., 2012; Sindelarova et al., 2014) Isoprene is a dialkene, and so is susceptible to oxidation in the atmosphere, initiated by the breaking of one, or both, of the double bonds.(Wennberg et al., 2018) Some of the products of

these reactions are organonitrates which are formed either by the reaction of isoprene with hydroxyl radicals (OH) and subsequent reactions with $O_2$ and NO, or by the addition of the nitrate radical ($NO_3$) to one of isoprene's double bonds. The resulting nitrates are important for their influence on the $NO_x$, $HO_x$, and $O_3$ budgets, as well as the potential for the formation of secondary organic aerosol (SOA) by condensation or via further reactions.(Emmerson and Evans, 2009; Bates and Jacob, 2019; Schwantes et al., 2019; Schwantes et al., 2020; Vasquez et al., 2020; Palmer et al., 2022)

This work focusses on three types of primary nitrates resulting from isoprene oxidation, and one group of secondary nitrates. The primary $C_5$ nitrates are the isoprene hydroxynitrates (IHN, Figure 1), isoprene carbonyl nitrates (ICN, Figure 2), and isoprene hydroperoxynitrates (IPN, Figure 3). The molecular formulae of IHN, ICN, and IPN are $C_5H_9NO_4$, $C_5H_7NO_4$, and $C_5H_9NO_5$, respectively. Throughout this work an upper-case sigma is used to denote the group of nitrates as well as any other species present in a chemical mechanism with the same molecular formula. For example, ΣIHN will refer to all

isoprene hydroxynitrates as well as any other $C_5H_9NO_4$ species present in each chemical mechanism. A glossary of the terms used to refer to different nitrated species is given in the supplementary information (Table S4).

IHN may be formed by OH-initiated oxidation followed by a peroxy radical ($RO_2$) + NO reaction, or by $NO_3$-initiated oxidation followed by $RO_2$ cross-reactions to form the alcohol group (Figure 1). ICN is formed by $NO_3$-initiated oxidation followed by $RO_2$ cross-reactions, hydrogen abstraction from alkoxy radicals (RO) by oxygen (RO + $O_2$ → ICN + $HO_2$), or

the reaction of IPN or isoprene dinitrates (IDN) with OH (Figure 2). IPN is formed by $NO_3$-initiated oxidation followed by $RO_2$ + $HO_2$ reactions (Figure 3).(Jenkin et al., 2015; Wennberg et al., 2018; Novelli et al., 2021; Vereecken et al., 2021)

The final group of nitrates are secondary nitrates with the formula $C_4H_7NO_5$, corresponding to the hydroxycarbonyl nitrate structures shown in Figure 4, which have been shown to be a major contributor to isoprene nitrates as measured by iodide chemical ionisation mass spectrometry ($I^-$-CIMS).(Tsiligiannis et al., 2022) $\Sigma C_4H_7NO_5$ refers to the isoprene-derived nitrates

as well as isomeric species present in the Master Chemical Mechanism (MCM) from other VOC sources.(Jenkin et al., 2015) There are several identified formation routes of $C_4H_7NO_5$ including the OH-initiated oxidation of methyl vinyl ketone (MVK) and methacrolein (MACR); $NO_3$-initiated oxidation of MVK and MACR; OH-initiated oxidation of IHN, IPN, and ICN; the ozonolysis of IHN; and the $NO_3$-initiated oxidation of hydroxycarbonyls (Figure 5).(Jenkin et al., 2015; Praske et al., 2015; Schwantes et al., 2015; Wennberg et al., 2018; Tsiligiannis et al., 2022) Analysis of these multifunctional

compounds is further complicated due to its secondary nature, as well as their potentially long atmospheric lifetime.(Müller et al., 2014)

Isoprene nitrates are often identified as major products of isoprene oxidation. For example, studies performed in the Forschungszentrum Jülich SAPHIR chamber identified a large range of organonitrates resulting from the $NO_3$-initiated oxidation of isoprene, including the primary products mentioned here.(Wu et al., 2021; Brownwood et al., 2021) Chamber

experiments performed at the California Institute of Technology have also highlighted the role of nitrates in the OH-initiated oxidation of isoprene.(Schwantes et al., 2019; Vasquez et al., 2020) Such nitrates have also been identified in a range of

ambient environments, from rural environments such as those in the south eastern United States, to polluted urban environments such as the San Francisco Bay area.(Ayres et al., 2015; Zaveri et al., 2020) Previous modelling studies that investigate isoprene nitrates under ambient conditions, and their impacts on atmospheric chemistry, are also widespread

across polluted and less polluted environments, examining both speciated nitrates and the sum of total organic nitrates.(Pratt et al., 2012; Xiong et al., 2015; Romer et al., 2016; Chen et al., 2018; Zare et al., 2018; Schwantes et al., 2020)

Isoprene nitrates have also been identified as significant species during the 2017 Atmospheric Pollution and Human Health in a Chinese Megacity (APHH) summer campaign in Beijing.(Hamilton et al., 2021; Newland et al., 2021) There have been two previous box-modelling investigations focussed on the data collected during the APHH-Beijing intensive field

observations.(Reeves et al., 2021; Whalley et al., 2021) Whalley *et al.* focussed on radical chemistry and ozone formation, highlighting several inconsistencies between modelled radical species and relevant measurements. Reeves *et al.* investigated IHN and ICN speciation and demonstrated the value of speciated measurements of isoprene nitrates by identifying several instances where the modelled IHN isomer distribution was not consistent with their measured distribution. They also discussed issues around the simplified representations of ICN isomers with regards to the initial site of attack of $NO_3$ and the

E/Z stereochemistry of 1,4-ICN and 4,1-ICN. This paper uses similar box-modelling approaches as the previously discussed studies to assess the capabilities of three detailed atmospheric oxidation mechanisms for investigating the formation and losses of isoprene derived nitrates in this anthropogenically and biogenically impacted environment. Key statistics for each mechanism are given in Table S1.

The first mechanism used here is the Master Chemical Mechanism v3.3.1 (MCM).(Jenkin et al., 2015) The MCM is a

benchmark near-explicit chemical mechanism extensively used by the atmospheric science community in a wide variety of science and policy applications where chemical detail is required. Subsets of the MCM can be directly extracted for a wide variety of VOCs (mcm.york.ac.uk). However, due to the breadth of the MCM, some simplifications have been made when constructing the mechanism. The first major simplification is the use of lumped $RO_2$ reactions. This means that $RO_2$-$RO_2$ cross-reactions are not treated explicitly, and it is assumed that each $RO_2$ will react with any other $RO_2$ at the same rate,

which helps to greatly reduce the complexity of mechanisms.(Jenkin et al., 1997) In the case of isoprene, further assumptions are made. For example, $NO_3$-initiated oxidation of isoprene in the MCM is represented by only one isomer (NISOPO2).

Secondly, the full v5 isoprene oxidation mechanism taken from the Wennberg *et al.* 2018 review of gas-phase isoprene oxidation (henceforth, the Caltech Mechanism) was used.(Wennberg et al., 2018) This mechanism treats isoprene $RO_2$ cross-reactions explicitly, unlike the lumped-$RO_2$ approach of the MCM. This leads to issues when integrating the Caltech

Mechanism with the MCM subset for additional measured VOCs, as explained further in the methodology section. The Caltech Mechanism aims to provide a more up-to-date representation of reaction rates and products. For example, the Caltech Mechanism provides four different nitrated $RO_2$ radicals resulting from $NO_3$ oxidation. The Caltech Mechanism also introduces some reactions that are not found in the MCM, such as intramolecular $RO_2$ reactions.

Finally, the mechanism developed by Vereecken *et al.* and further expanded in Tsiligiannis *et al.* was used and is referred to

as the FZJ Mechanism.(Vereecken et al., 2021; Tsiligiannis et al., 2022) This mechanism aims to expand on the Caltech

Mechanism, by providing more comprehensive $NO_3$ chemistry, including the proposed formation of epoxide species from some alkoxy radical species, and additional chemistry relevant to $C_4H_7NO_5$ outlined in Tsiligiannis *et al.*(Tsiligiannis et al., 2022)

## 2 Methodology

### 2.1 Ambient Measurements

The Beijing measurements used in this work were collected at ground level at the Tower Section of the Institute of Atmospheric Physics (IAP) in Beijing, China, between 2017-06-01 and 2017-06-18.(Shi et al., 2019) The nitrates were measured using a Filter Inlet for Gases and Aerosols (FIGAERO) coupled to a time-of-flight iodide chemical ionisation mass spectrometer ($I^-$-CIMS) which allows for the measurement of particle and gas-phase species, although only the gas-phase data are used here as the particle-phase data were unavailable.(Lopez-Hilfiker et al., 2014) Each nitrate was calibrated assuming the same sensitivity as trans-beta-IEPOX, though the potential role of calibration on the measured nitrate concentrations is discussed throughout this work.(Hamilton et al., 2021) Other organic compounds were measured by proton transfer mass spectrometry (PTR-MS), selected ion flow tube mass spectrometry (SIFT-MS), and dual-channel gas chromatography with flame ionization detection (DC-GC-FID).(Hopkins et al., 2011; Huang et al., 2016; Shi et al., 2019; Reeves et al., 2021) The sum of monoterpenes measured by PTR-MS and SIFT-MS was used to constrain alpha-pinene and limonene in the models, assuming each compound comprised 50% of the total monoterpenes. Instruments used to measure organic species are summarised in Table S2 and the details of the instruments used to measure additional compounds can be found elsewhere.(Whalley et al., 2010; Whalley et al., 2018; Zhou et al., 2018; Shi et al., 2019; Hamilton et al., 2021; Whalley et al., 2021) Where species constraints were required in the modelling, and multiple measurements were taken, the mean of all of the measurements was used. The scanning mobility particle sizer (SMPS) instruments used to calculate particle surface area as outlined in Section 2.3.1 are described in the Supplementary Information.

### 2.2 Mechanisms

This investigation involved a comparison of three different isoprene oxidation mechanisms. The MCM subset for isoprene and the additional VOCs which were measured throughout the campaign and were available in the MCM (Table S2) was extracted directly from the MCM website (mcm.york.ac.uk).(Jenkin et al., 2015) The MCM inorganic chemistry scheme was used for all three mechanisms.

The Caltech Mechanism was integrated with the MCM subset for the additional VOCs by producing lumped $RO_2$ cross-reactions using the approach outlined in Jenkin *et al.*(Jenkin et al., 1997)  For each $RO_2$ species where explicit reactions are given, the geometric mean of the self-reaction rate and the $CH_3O_2$ self-reaction rate was used. If a self-reaction was not specified, then the $CH_3O_2$ self-reaction rate was used. Branching ratios were then applied to the alcohol-forming, carbonyl-forming, and alkoxy-forming reactions according to Jenkin *et al.*

The FZJ Mechanism was produced by adding the reactions outlined in Tsiligiannis *et al.* to the mechanism provided in Vereecken *et al.* and combining it with the MCM subset for measured non-isoprene species. (Vereecken et al., 2021; Tsiligiannis et al., 2022)

Each of the mechanisms used in this work have been made available online (doi.org/10.15124/500474f7-6e69-47db-baf7-36310451fd15).

## 2.3 Modelling Approach

AtChem2, an open-source zero-dimensional box-model tool, was used in this work.(Sommariva et al., 2019) A separate model was run for each day to avoid compounding errors carrying across multiple days of the model, for example the uncertainty that may result from imperfect accounting for physical processes. $NO_2$, $O_3$, CO, $SO_2$, HONO, and formaldehyde, along with 29 primary VOCs for which data were available (Table S2), were all constrained to the 30-minute averaged measured values throughout the campaign. NO was left unconstrained due to the potential for local NO emissions to result in mixing ratios unrepresentative of the larger area that is important for the formation of long-lived organic products such as organonitrates. Constraining to NO would result in unrealistically low $NO_3$ concentrations by increasing the rate of the $NO_3$ + NO reaction based on elevated NO concentrations. Temperature, pressure, boundary-layer height, and relative humidity were also constrained to measured values. Photolysis values in the models were constrained to measured values where available ($J_{O1D}$, $J_{NO2}$, $J_{HONO}$, $J_{HCHOr}$, $J_{HCHOnr}$, $J_{NO3toNO}$, $J_{NO3toNO2}$, $J_{CH3CHO}$, $J_{CH3OCH3}$), and remaining photolysis rates were calculated according to the parameterization used in the MCM and scaled based on the ratio of the calculated and measured $J_{NO2}$. The models consisted of a 24-hour spin-up period followed by a further 24-hour period. Constraints were made by duplicating the measured values for each day to provide a 48-hour constraint of two repeated 24-hour periods. The model output was then considered to be the model output in the second 24-hour period of the model run. The model outputs were then concatenated to produce a time series across the whole period of interest.

To account for the deposition of species to surfaces, deposition reactions were added for all species. Each species was assigned a deposition velocity based on the functionality of that compound. Deposition velocities for $H_2O_2$, $HNO_3$, and $O_3$ were applied directly to each compound. Separate deposition velocities for organic hydroperoxides and organic nitrates were applied to compounds containing the hydroperoxide and nitrate functional groups. Organic acid species were assigned the formic acid deposition velocity, and a general oxidised VOC deposition was assigned to carbonyl and alcohol containing compounds. The rate of deposition was determined by dividing the assigned deposition velocity by the measured boundary layer height. All deposition velocities were taken from Nguyen *et al.* 2015 and are summarised in Table S3.(Nguyen et al., 2015) For multifunctional compounds, the largest deposition velocity of each of the functional groups present in the compound was selected from Table S3.

Additionally, a loss term was included for all species to account for mixing and ventilation. A diurnally varying ventilation rate was applied, where the rate was scaled such that the modelled glyoxal concentrations matched measurements, in a

similar fashion to previous work. (Whalley et al., 2021; Reeves et al., 2021) The sensitivity of the model results to this term is assessed in the Model Validation section.

### 2.3.1 Particle Phase Processes

In the cases of $\Sigma$IHN and $\Sigma$IPN, an analysis of the impact of the particle-phase hydrolysis of 1,2-IHN and the reactive uptake of INHE is performed. For both of these cases, the rates of loss ($k_{IHN}$ and $k_{IHNE}$ for IHN hydrolysis and INHE uptake respectively) are calculated using Equation 1. $S_a$ is the aerosol surface area, as calculated for each model time-step from scanning mobility particle sizer (SMPS) measurements, $r_p$ is the effective particle radius calculated as a weighted median of the SMPS number measurements at each model time-step, $D_g$ is the gas-phase diffusion coefficient, $\nu$ is the mean molecular speed of IHN or INHE molecules in the gas phase, and $\gamma$ is the reactive uptake coefficient. $\nu$ was calculated using Equation 2 where R is the ideal gas constant (8.314 J K$^{-1}$ mol$^{-1}$), T is the measured temperature at each time-step, and $M_r$ is the molecular mass of the compound of interest (0.147 kg mol$^{-1}$ for IHN and 0.163 kg mol$^{-1}$ for INHE). A value of $1\times10^{-5}$ m$^2$ s$^{-1}$ was used for $D_g$, as is assumed in Gaston *et al.* for IEPOX. (Gaston et al., 2014) This method has been extensively used to calculate the rate of reactive uptake of IEPOX. (Gaston et al., 2014; Riedel et al., 2016; Budisulistiorini et al., 2017)

$$k_{IHN} = \frac{S_a}{\frac{r_p}{D_g} + \frac{4}{\nu\,\gamma_{IHN}}} \qquad\qquad Equation\ 1$$

$$\nu = \sqrt{\frac{3\,R\,T}{M_r}} \qquad\qquad Equation\ 2$$

An estimation of $\gamma$ is complicated by the dependence on particle properties. In each case, results are shown for models where a range of $\gamma$ values are assumed, between the limits of 0 and 1.

## 3 Results and Discussion

### 3.1 Model Validation

When comparing the measured and modelled NO mixing ratios, there is good agreement during the day-time, with the models deviating from the measurement by a maximum of around 2 times (Figure 6a). The models do not reproduce the elevated night-time NO concentrations observed in Beijing, however this night-time NO is likely the result of local emissions and so will have little impact on the chemistry that is the focus of this study. Figure S1 shows the good match between modelled NO and NO measured at an altitude of 100m showing the ability of the model to predict NO away from local sources. This is further confirmed by NO$_3$ predictions provided by the models being, at most, 2.5 times over-predicted (Figure 6b). There is also a slight under-prediction of NO$_3$ by a factor of around 0.4 during the afternoon.

HO$_x$ predictions from the models are generally good. There is close agreement to the measured OH concentrations, although the modelled concentrations are around 0.5 times the measured values during the morning period (Figure 6c). Day-time HO$_2$

concentrations are around 2 times higher than the measurement during the evening in all models (Figure 6d), which is consistent with findings from Whalley *et al.* 2021 where a similar box-model run using the MCM over-predicted $HO_2$, particularly during low-NO periods. Whalley *et al.* hypothesises that the $HO_2$ over-prediction may be caused by unaccounted for RO isomerisation reactions that result in $RO_2$ radical formation without concurrent $HO_2$ formation.(Whalley et al., 2021) While the Caltech Mechanism and FZJ Mechanism both include additional RO isomerisation reactions for isoprene, they inherit the MCM RO chemistry for other VOCs, including longer-chain VOCs that may be more susceptible to RO isomerisations, and so this could still be a reasonable hypothesis. The major contributors to RO composition in the models are aromatic species owing to their relatively long lifetimes.

When comparing the modelled and measured MVK and MACR mixing ratios, while day-time concentrations are at-most half of the measured values, the night-time concentrations fall far below the measurements (Figure S2). This may be the result of the long lifetime of MVK and MACR, meaning there is a high background concentration not captured by the models. Alternatively, it may due to imperfect accounting for physical processes such as mixing and ventilation within the models or a poor understanding of MVK+MACR chemistry in this environment. There may also be some role played by the conversion of isoprene hydroxyhydoperoxides to MVK+MACR on the metal inlets of the mass spectrometers resulting in an artificially increased measurement. (Rivera-Rios et al., 2014; Newland et al., 2021) It is also important to consider the effect of upwind isoprene concentrations for all of the isoprene oxidation products discussed in this work. While our modelling makes use of isoprene concentrations measured at the same site as the product measurements, the upwind isoprene concentrations would be more useful for predicting the concentrations of isoprene oxidation products.

While a ventilation term is included in the models, and is scaled to glyoxal concentrations, there is uncertainty as to its true rate and diurnal variability. As a test of the models' sensitivity to the ventilation rate, the rate was halved and doubled in two separate tests (Figure S3). The halving of the ventilation rates resulted in an average change in concentration across the models run with each mechanism of 3.1, 1.5, 1.8, and 1.8 times for $\Sigma C_4H_7NO_5$, $\Sigma IHN$, $\Sigma ICN$, and $\Sigma IPN$ respectively. The average changes for doubling the ventilation rate were 0.32, 0.62, 0.60, and 0.56 for $\Sigma C_4H_7NO_5$, $\Sigma IHN$, $\Sigma ICN$, and $\Sigma IPN$ respectively. Xiong *et al.* aimed to reduce the impact of ventilation by analysing nitrates as ratios with the sum of MVK and MACR.(Xiong et al., 2015) However, due to the differences in MVK+MACR predicted using each mechanism, using the MVK+MACR ratio as a proxy for the absolute concentration of the nitrates complicates the comparison of different mechanisms. As such, the analysis here involves the use of mixing ratios as opposed to the ratios relative to MVK+MACR. In order to analyse the average trends over a day within the modelled period, average diurnal plots are used to examine the modelled and measured data. The mean diurnals are used here, though use of the median had little impact on the diurnal values.

Comparison of the MVK+MACR predicted using each mechanism is consistent with the work presented in Vereecken *et al.* (Vereecken et al., 2021) Figure S2 shows that the Caltech Mechanism produces the highest night-time MVK+MACR concentrations with the MCM and FZJ Mechanism producing the lowest night-time concentrations. The MCM does not include MVK+MACR formation from isoprene+$NO_3$ chemistry, while the Caltech Mechanism does. The FZJ Mechanism

does include some MVK+MACR formation from isoprene $NO_3$ chemistry, but also reduces the yield from ozonolysis reactions resulting in similar MVK+MACR yields between the MCM and FZJ Mechanism in Vereecken *et al.* and in the night-time period of the models presented here. During the day-time, the FZJ models produce the lowest MVK+MACR concentrations as this adjusted ozonolysis chemistry becomes more significant.

  Isoprene epoxydiols (IEPOX) are a significant contributor to isoprene-derived SOA and are significant isoprene oxidation
products along with the isobaric isoprene hydroxyhydoperoxides (ISOPOOH).(Paulot et al., 2009; Surratt et al., 2010; Nguyen et al., 2014) Figure S4 shows the modelled and measured ΣIEPOX+ISOPOOH. All three mechanisms resulted in a large under-prediction of ΣIEPOX+ISOPOOH. As with MVK+MACR, this under-prediction may result from ventilation from the model being too rapid. As discussed throughout the manuscript, there may also be an issue of calibration for the $I^-$-CIMS data. Although the $I^-$-CIMS data is calibrated using IEPOX, all three models predict around half of the
ΣIEPOX+ISOPOOH to be comprised of ISOPOOH. Accounting for particle-uptake of IEPOX would only increase this fraction of ISOPOOH. Additionally, there are multiple IEPOX isomers whereas this data is calibrated to only one isomer. More discussion of calibration issues is given in Section 3.2.1.

  The volatility of the nitrate species was assessed in order to determine the potential impact of condensation to the particle phase. An equilibrium partitioning approach was taken, as described in Mohr *et al.* 2019.(Mohr et al., 2019) This resulted in
common logarithm of saturation concentrations in units of molecules $cm^{-3}$ ($\log(C_{sat})$) of between 4.0 and 5.3, revealing the high volatility of these compounds. As such, the condensation of these nitrates to the particle phase is assumed to be negligible, though this approach does not account for reactive uptake to particles.

### 3.2 ΣIHN ($C_5H_9NO_4$)

  Throughout the day, the three mechanisms produce similar ΣIHN mixing ratios, at approximately half of the measured value
(Figure 7). Despite the absolute differences, the profile of modelled ΣIHN matches the measurement, with decreasing mixing ratios in the afternoon reflecting the titration of NO by increasing $O_3$. (Newland et al., 2021) Reeves *et al.* shows reasonable predictions of the major IHN isomer (1,2-IHN) made by their MCM-based model, whereas the modelled 4,3-IHN showed an over-prediction of around two times at mid-day.(Reeves et al., 2021) This discrepancy is likely the result of different representations of physical processes in the models. The time series for modelled and measured ΣIHN is shown in Figure S5.
Figure 8 shows the clear split between the day-time and night-time IHN speciation in all of the models. Figure 8 also demonstrates that the contribution of non-IHN species to ΣIHN in the models is very small, meaning a measured ΣIHN ($C_5H_9NO_4$) signal is likely to be a reasonable measurement of IHN. Both OH and $NO_3$ addition to isoprene favours the terminal carbon atoms, so OH oxidation followed by reaction with NO results in the nitrate group being formed either on one of the central positions or the remaining terminal carbon. This means OH-initiated oxidation predominantly forms 1,2-IHN,
4,3-IHN, E/Z-1,4-IHN, and E/Z-4,1-IHN. $NO_3$ addition results in the nitrate group being present on the terminal carbons, at the initial site of attack.(Wennberg et al., 2018) This means $NO_3$-initiated oxidation predominantly forms 2,1-IHN, 3,4-IHN, E/Z-1,4-IHN, and E/Z-4,1-IHN.

The night-time shows an enhancement in IHN species produced by $NO_3$ chemistry. This is most obvious in the MCM model, where all isoprene + $NO_3$ chemistry is channelled through just one isomer, ISOPCNO3. As such, ISOPCNO3 makes up very little of the day-time IHN, but up to 80% of night-time IHN just before sunrise. Similarly, the ΣIHN modelled using the Caltech Mechanism and FZJ Mechanism are almost exclusively comprised of ISOP1OH2N and ISOP3N4OH during the day, but there is a more even distribution at night with major contributions from ISOP1N2OH, ISOP1N4OHt, and ISOP1N4OHc. The FZJ Mechanism contains a reduced rate of ISOP1N2OH formation from ISOP1N2OO cross-reactions compared to the Caltech Mechanism, hence the lower contribution of 'NO₃-initiated IHN' to ΣIHN in the FZJ Mechanism model.

Previous work has shown that the hydrolysis of 1,2-IHN occurs rapidly in the atmosphere.(Vasquez et al., 2020; Liu et al., 2012) To test the sensitivity of our results to 1,2-IHN hydrolysis, loss reactions of 1,2-IHN were added to each of the mechanisms with a rate calculated as described in Section 2.3.1. Figure S6 shows the modelled ΣIHN using each of the mechanisms with 1,2-hydrolysis reactions included. Since the majority of daytime ΣIHN is comprised of 1,2-IHN, removal of this compound can have a large effect on the modelled ΣIHN. A $\gamma_{IHN}$ value of 1 removes most, but not all, of the 1,2-IHN and a value of 0.1 brings modelled ΣIHN concentrations close to when the value is 1. Conversely, $\gamma_{IHN}$ values below 0.01 only result in small changes to modelled ΣIHN compared to the base model where no IHN hydrolysis is included.

### 3.2.1 ΣIHN Calibration

As previously noted, the I⁻-CIMS data presented here is calibrated relative to IEPOX, which results in two potential issues. Firstly, the sensitivity of I⁻-CIMS to the compounds of interest may be significantly different from the sensitivity to IEPOX, leading to a bias in the measurement. Secondly, if I⁻-CIMS has different sensitivities to the different isomers of a particular formula, the changing isomer distribution over time will result in a varying sensitivity to the entire m/z signal as each isomer contributes more or less. For example, it has been previously shown that I⁻-CIMS is more sensitive to IHN isomers in which the NO3 group is located close to the OH group, such as 4,3-IHN and Z-1,4-IHN. Isomers where the NO3 and OH groups are not in close proximity, such as E-1,4-IHN, show much lower responses to iodide-adduct ionisation. (Lee et al., 2014) The "Mixed-source IHN" in Figure 8 includes both E and Z isomers of 1,4-IHN and 4,1-IHN. Since there is a higher proportion of mixed-source IHN during the night in all models, the sensitivity of ΣIHN can be expected to be lower at night than during the day due to a higher proportion of E-1,4-IHN and E-4,1-IHN.

Lee *et al.* report sensitivity values for IEPOX alongside the sensitivity values for three IHN isomers (4,3-IHN, Z-1,4-IHN, and E-1,4-IHN).(Lee et al., 2014) Dividing the sensitivities of each of these isomers by the IEPOX sensitivity allows a relative sensitivity to be obtained for each. These relative sensitivities are 15.64, 14.62, and 0.9487 for 4,3-IHN, Z-1,4-IHN, and E-1,4-IHN respectively. Relative sensitivities for the remaining IHN isomers can be assigned based on the orientation of the OH and NO₃ groups.(Xiong et al., 2015) A total ΣIHN sensitivity can then be estimated using the modelled isomer distribution from each set of models. Figure 9a shows the diurnally varying relative sensitivity for each of the models. The largest discrepancy between the models can be seen at night, resulting from the differing $NO_3$ chemistry in each mechanism.

Taken together, the models indicate that I⁻-CIMS may be between 2.5 to 1.4 times less sensitive to $\Sigma$IHN during the night than during the day.

Applying this relative $\Sigma$IHN sensitivity to the IEPOX calibrated data dramatically reduces the measured concentrations of $\Sigma$IHN, due to the high sensitivities of the majority of IHN isomers (Figure 9b). It is interesting to note differing $\Sigma$IHN concentrations predicted using the isomer distribution from each mechanism. At midnight, the FZJ-adjusted $\Sigma$IHN data is around twice that of the Caltech-adjusted data. According to this adjusted $\Sigma$IHN data, all of the models would be over-predicting $\Sigma$IHN by around an order of magnitude. Even when comparing to the most extreme 1,2-IHN hydrolysis case previously presented, $\Sigma$IHN concentrations are over-predicted by 1.5 to 3 times compared to the adjusted I⁻-CIMS data. Additionally, the adjusted calibration factors change the shape of the $\Sigma$IHN diurnal, resulting in a second peak in mixing ratios at around 20:00. Using the isomer distribution predicted by the FZJ mechanism suggests that this second night-time peak could be as large as the mid-day peak.

The use of relative responses here aims to eliminate some issues associated with the direct comparison of data from different instruments, but may not eliminate all of the unknown differences. Nevertheless, adjusting the measured $\Sigma$IHN in this way suggests that the perceived under-prediction in $\Sigma$IHN by all of the models may instead be a closer representation to the true $\Sigma$IHN concentrations, if not an over-prediction. IHN is the most widely studied of the nitrates presented here and so the calibration correction can be applied quantitatively, however the impact of calibration on the measured organonitrate concentrations must be considered throughout this work.

### 3.3 $\Sigma$IPN ($C_5H_9NO_5$)

The measured $\Sigma$IPN shows little diurnal variation (Figure 10). Contrary to observations, all models produced strong diurnal profiles of $\Sigma$IPN. This is because the majority of IPN is formed through $NO_3$ oxidation of isoprene at night when there are few losses. The only losses of IPN in all mechanisms, besides the added deposition reactions, are photolysis reactions and the reaction with OH. The strong diurnal profile results in night-time mixing ratios being over-predicted by around 1.5 times and day-time mixing ratios being close to 0. Both the MCM and FZJ Mechanism result in $\Sigma$IPN reaching a minimum at sunrise, slightly increasing throughout the day, before a rapid night-time increase. The daytime under-prediction of $\Sigma$IPN may be indicative of mixing in the models being overestimated. The time series for modelled and measured $\Sigma$IPN is shown in Figure S7. The data presented in Figure S7 show that there is substantial noise in the $\Sigma$IPN data, which may also mask diurnal trends and indicates that the $\Sigma$IPN concentrations are close to the instrument's detection limit for these compounds.

While none of the mechanisms include $NO_3$ or $O_3$ oxidation of IPN, the Wennberg *et al.* 2018 review of isoprene chemistry does list estimated reaction rates of IPN, ICN, and IHN with $NO_3$, $O_3$, and OH. (Wennberg et al., 2018) Figure S8 shows the average proportional night-time chemical loss for IHN, IPN, and ICN calculated using the rates given in Wennberg *et al.* and the measured OH, $O_3$, and $NO_3$ concentrations between 20:00 and 05:00. For the IPN isomers, OH oxidation accounts for the majority of the chemical loss of IPN at night, with around 10-15% being lost to reaction with $NO_3$. Reaction with $O_3$ also makes up a substantial fraction of the chemical loss in the 1,4-IPN and 4,1-IPN isomers, though OH is still the major sink.

Since OH oxidation is included in the mechanisms, then the majority of the chemical losses should be captured by the models. Physical processes also dominate the losses of $\Sigma$IPN at night, so the addition of more chemical losses would not have a large impact on $\Sigma$IPN concentrations.

To understand the trends in $\Sigma$IPN, it is important to consider the multiple isomeric (non-IPN) species present in each of the mechanisms which can make up a large proportion of the modelled $\Sigma$IPN (i.e. species with the formula $C_5H_9NO_5$). The most significant isomers of IPN are C51NO3, originally from the MCM and present in all mechanisms, C524NO3, originally from the MCM and also present in the FZJ mechanism, ISOP1N23O4OH, present in the Caltech Mechanism and FZJ Mechanism, and ISOP1N253OH4OH, present in the Caltech Mechanism (Figure S9).

C51NO3 is a nitrated hydroxy carbonyl compound in the MCM with formation routes from isoprene, as well as from hydrocarbons such as pentane. C524NO3 is an isoprene OH oxidation product from the MCM. In the MCM and FZJ Mechanism models, C51NO3 and C524NO3 make up the majority of modelled $\Sigma$IPN composition during the day-time (Figure S10). These are the species responsible for the slight increase in $\Sigma$IPN throughout the day in the MCM and FZJ Mechanism models. C51NO3 and C524NO3 production from isoprene is not included in the Caltech Mechanism, and the only formation routes to C51NO3 are from non-isoprene species. As such, C51NO3 and C524NO3 only makes a small contribution to total $\Sigma$IPN in the Caltech Mechanism model and the day-time increase is not present.

ISOP1N253OH4OH is only present in the Caltech Mechanism and is initially formed from an intramolecular H-shift of the 1,4 isoprene alkoxy nitrate (INO), ISOP1N4O. The Caltech Mechanism does not contain any loss reactions for this species, which may account for its moderate contribution to modelled night-time $\Sigma$IPN (Figure S10). This INO H-shift pathway is not included in the FZJ Mechanism and so ISOP1N253OH4OH is not present.

ISOP1N23O4OH is a nitrated hydroxyepoxide that was proposed, alongside other positional isomers which are produced by the models in lower amounts, as a product of IPN OH oxidation by Schwantes *et al.* where it is termed isoprene nitrooxy hydroxyepoxide (INHE).(Schwantes et al., 2015) While the formation of INHE from IPN is present in the Caltech Mechanism, epoxidation reactions from alkoxy radicals that are predicted in Vereecken *et al.* result in much more INHE production in the FZJ Mechanism model. The FZJ Mechanism model results predict that at midnight, around half of the total $\Sigma$IPN is composed of INHE (Figure 11). If such large concentrations of these epoxides are produced, then this could have a significant impact on SOA formation via reactive uptake in a similar fashion to IEPOX.(Paulot et al., 2009; Surratt et al., 2010; Schwantes et al., 2015; Hamilton et al., 2021)

In order to assess the potential for reactive uptake of INHE on the modelled $\Sigma$IPN, loss reactions for each of the four INHE isomers in the FZJ Mechanism were added to the mechanism and the models rerun. The rate coefficient for the reactive uptake of INHE ($k_{INHE}$) was calculated as described in Section 2.3.1. Figure S11 shows the modelled $\Sigma$IPN produced by a set of models for which a range of $\gamma_{INHE}$ were assumed, between the limits of 0 and 1. When $\gamma_{INHE}=1$ and $\gamma_{INHE}=0.1$, almost all of the INHE is removed from the gas-phase at any time which brings the modelled night-time concentrations of $\Sigma$IPN to around two thirds of the measured value. When $\gamma_{INHE} = 0.01$, the modelled night-time $\Sigma$IPN is reasonably in line with the measurements between 20:00 and 00:00, after which the modelled concentrations fall with the diurnal profile explained

previously. $\gamma_{INHE} = 0.001$ results in modelled concentrations close to the values without any particle uptake. Previous estimations of the reactive uptake coefficient of IEPOX ($\gamma_{IEPOX}$) usually range between $7\times10^{-2}$ and $2\times10^{-4}$, though measurements have been made as low as $9\times10^{-7}$. (Gaston et al., 2014; Riedel et al., 2015; Budisulistiorini et al., 2017)

As with all of the nitrates investigated here, the role of the I⁻-CIMS calibration on the data presented must be considered. As shown previously, all models predict a diurnally varying isomer distribution with night-time ΣIPN being largely comprised of IPN and/or INHE, and daytime ΣIPN being comprised of smaller concentrations of other species. If the daytime isomers were much more sensitively detected than the night-time isomers then this could offset the diurnal concentration profile modelled to produce a constant measured signal throughout the day, as is observed. The daytime ΣIPN concentrations predicted by the MCM and FZJ models is around 0.06 times the measured values, meaning that the daytime isomers would need to be around 17 times more sensitively detected than IEPOX to reproduce the flat diurnal signal observed, assuming the night-time isomers had the same sensitivity as IEPOX. There has been very little research to quantify the sensitivity of I⁻-CIMS to hydroperoxides, but Lee *et al.* reported the sensitivity of peroxyacetic acid to be 0.04 times that of acetic acid suggesting that the non-hydroperoxide daytime nitrates may be more sensitively detected than the night-time IPN. (Lee et al., 2014)

### 3.4 ΣICN ($C_5H_7NO_4$)

ΣICN shows the largest difference between mechanisms. In line with the measurements, all models show low concentrations of ΣICN during the day (Figure 12). ΣICN then increases at sunset, due to $NO_3$-initiated formation from isoprene, and then reduces in concentration into the early morning as production ceases. There is a large over-prediction of a factor of around 25 times in the night-time mixing ratio modelled using the MCM which is consistent with findings from Reeves *et al.* who also found ICN to be over-predicted in their models using the MCM, however the lack of NO constraint in our models results in slightly higher modelled ICN concentrations due to elevated $NO_3$ concentrations, hence the discrepancy between the model and measurement is slightly larger in this work.(Reeves et al., 2021) This over-prediction decreases to around 7 times when using the Caltech Mechanism, and decreases further to around 3 times when using the FZJ Mechanism. A plot of ΣICN concentrations normalised to the concentration at midnight is shown in Figure S12. The time series for measured and modelled ΣICN is given in Figure S13.

The large over-prediction made by the MCM is the result of large production terms from the decomposition of all INO radicals (represented by NISOPO in the MCM) into ICN. In contrast, the Caltech Mechanism provides alternative INO decomposition routes including fragmentation and H-shift autoxidation reactions (Figure S14). The FZJ Mechanism includes much of this updated chemistry as well as proposing the previously discussed epoxide formation reactions from some alkoxy radicals, which further reduces the ICN production route (Figure S14). The improvement in predictions of ΣICN indicates that the assumption made by the MCM of 100% of INO decomposing to form ICN is unlikely to be valid. The loss of ΣICN is dominated by physical processes in all of the models, particularly at night when ΣICN concentrations are the highest. Additional ICN losses being added to the MCM may improve ΣICN predictions, for example Hamilton *et al.* proposed ICN

as a precursor to particle-phase species observed in Beijing via an isoprene nitrooxy hydroxy-α-lactone (INHL) species.(Hamilton et al., 2021) However, the MCM already includes reactions with $O_3$ and $NO_3$ that are not included in the Caltech or FZJ Mechanisms, suggesting that the issue lies in the MCM's faster formation processes. Further discussion of the uncertainties in ICN losses is given by Reeves *et al.*(Reeves et al., 2021)

While this account of increasingly complex alkoxy radical chemistry gives good reason to question the high ICN formation rates from the MCM, it is also important to consider that previous work has found the lower sensitivity to aldehyde and ketone groups by I$^-$-CIMS compared to alcohols, as such it should be expected that the measured ΣICN is most likely to be under-quantified by use of the IEPOX calibrant compared to species such as IHN.(Lopez-Hilfiker et al., 2014; Iyer et al., 2016; Lee et al., 2014) For example, Lee *et al.* 2014 shows that the sensitivity to hydroxyacetone is around 20 times lower

than the similarly structured 1,2-butanediol and the sensitivity to 2,5-hexanedione is around 70 times lower than that of 5-hydroxy-2-pentanone. Assuming the relative sensitivity of ICN to IEPOX is lower than that of IHN, i.e. the sensitivity relative to IEPOX is lower than 15.64 (Section 3.2.1), would mean that the over-prediction made by the MCM could not be solely accounted for by the calibration. However, it is more difficult to comment on the accuracy of the FZJ mechanism compared to the Caltech mechanism in this respect as a reasonable calibration correction could bring the measurement in line

with either model.

### 3.5 $\Sigma C_4H_7NO_5$

$\Sigma C_4H_7NO_5$ mixing ratios are under-predicted by around an order of magnitude in all models (Figure 13). The modelled $\Sigma C_4H_7NO_5$ diurnals only slightly vary between each model, despite the additional dark formation rates added to the FZJ mechanism, with the Caltech mechanism actually producing the highest concentrations. This is because the formation of

410 $\Sigma C_4H_7NO_5$ is dominated by the OH oxidation of MVK and MACR. The time series for measured and modelled $\Sigma C_4H_7NO_5$ is given in Figure S15.

The under-prediction in MVK+MACR and the potentially high ventilation (see Section 3.1) may account for some of this under-prediction, particularly in light of the potentially long lifetime of $C_4H_7NO_5$, however the under-prediction is much stronger than is observed for the MVK+MACR precursors. (Müller et al., 2014) Without previous work investigating the

415 sensitivity of I$^-$-CIMS to $C_4H_7NO_5$ it is difficult to assess the impact of calibration on this measurement. Assuming a similar sensitivity as the most sensitively detected IHN isomer, where the OH and $NO_3$ groups are in close proximity like in the $C_4H_7NO_5$ isomers, would bring the measurement in line with the models.

### 4 Conclusions

Model results have been presented making use of three different detailed chemical mechanisms, comparing their predictions

of several isoprene organonitrates. While the gas-phase box-modelling approach used here allows for the use of such complex mechanisms, the simplified representation may not fully represent physical processes such as boundary layer

mixing in the morning and evening. Additionally, hydrolysis and aerosol uptake processes are not included in the mechanisms, meaning there may be unaccounted losses for species such as INHE. While the impact of I⁻-CIMS sensitivity on measurements of these nitrates has been considered throughout this work, the availability of authentic standards would greatly improve the ability to quantify such organonitrates.

When considering $\Sigma$IPN, the model results presented here indicate that large proportions of the measured $\Sigma$IPN can be composed of non-IPN species. This is especially true during the day-time, when $\Sigma$IPN concentrations are lowest. However, the epoxide-forming reactions proposed by Vereecken *et al.* suggest that around half of the measured night-time $\Sigma$IPN could be comprised of INHE.(Vereecken et al., 2021) Assuming reactive uptake coefficients similar to those previously measured for IEPOX results in small reductions in predicted $\Sigma$IPN, meaning that the FZJ mechanism predicts $\Sigma$IPN to be comprised of mostly non-IPN species for the majority of the day. Further studies of isoprene nitrate chemistry should investigate these species with techniques able to distinguish between the isomeric $\Sigma$IPN compounds and their reaction products, such as chromatographic techniques, in order to determine the role of INHE in isoprene oxidation. Such large INHE production terms would have implications for the formation and growth of secondary organic aerosol (SOA) by reactive uptake to acidified particles.(Hamilton et al., 2021) Generally, the large contribution of non-IPN species to the modelled $\Sigma$IPN highlights the caution that should be applied in interpreting measurements of $\Sigma$IPN solely as a measurement of IPN.

The changing distribution of $\Sigma$IHN isomers over the course of 24-hours has implications for the calibration of $\Sigma$IHN measurements. For example, I⁻-CIMS could be 2.5 to 1.4 times less sensitive to $\Sigma$IHN overnight where $NO_3$ chemistry is dominant, due to the increased contribution of E-1,4-IHN and E-4,1-IHN to $\Sigma$IHN. This means that the use of a constant calibration factor is likely to under-quantify night-time IHN, even if the calibration factor was accurate during the day. Furthermore, while comparison of the models to IEPOX-calibrated data suggests an under-prediction by the models, adjusting this calibration to account for the sensitivity of IHN isomers suggests a potentially very large over-prediction by the models.

The much improved $\Sigma$ICN predictions when using the Caltech and FZJ Mechanisms compared to the MCM indicates that the assumptions around alkoxy radical decomposition made by the MCM are likely to be inaccurate, even when calibration uncertainties are accounted for. Future studies focussed on isoprene nitrates should not overlook the inclusion of more complex INO decomposition routes, beyond the direct decomposition route to ICN present in the MCM.

While the results presented here surrounding $C_4H_7NO_5$ are not conclusive, there is potential for all of the mechanisms to be under-predicting $C_4H_7NO_5$. Additional $C_4H_7NO_5$ from $NO_3$ chemistry, as is included in the FZJ Mechanism model, does not improve predictions as the majority of the modelled $C_4H_7NO_5$ resulted from OH chemistry. Assuming an I⁻-CIMS sensitivity of $C_4H_7NO_5$ similar to that of the more sensitively detected IHN isomers would mean that the modelled $C_4H_7NO_5$ is approximately correct.

While physical processes dominated the loss of the organonitrates in all of the models presented here, the chemical losses of these species are not well understood. Estimated rate constants for the reaction of IHN, IPN, and ICN from Wennberg *et al.* indicate that the OH reactions which are included in all of the mechanisms may be the major chemical loss pathways, with

NO$_3$ oxidation comprising a larger loss than reaction with O$_3$. This has implications for NO$_x$ recycling, indicating that most of the NO$_x$ consumed to form the organonitrates is subsequently lost from the gas-phase or transported away from the site of formation. (Bates and Jacob, 2019)

Generally, the mechanisms presented here do a reasonable job at reproducing isoprene nitrate chemistry in Beijing, particularly with the inclusion of improved alkoxy radical chemistry, though it is clear that better constraints on the sensitivity of I$^-$-CIMS to nitrated compounds would aid in the analysis of these compounds.

**Author Contributions**

A.W.M performed the model simulations and prepared the manuscript. P.M.E and J.F.H provided supervision and advice throughout the project. B.H.L and J.A.T provided advice and feedback on the representation of isoprene nitrates in box-models and their measurement. A.R.R and B.S.N assisted in constructing the modelling approach and advised on the use of chemical mechanisms. T.J.B, J.B, J.R.H, J.D.L, C.P, and M.D.S carried out the measurements of species used for model constraints. All authors provided feedback on early drafts of the manuscript.

**Competing interests**

The authors declare that they have no conflict of interest.

**Acknowledgements**

The authors acknowledge the support from Pingqing Fu, Zifa Wang, Jie Li, and Yele Sun from IAP for hosting the APHH Beijing campaign at IAP. They also thank Tuan Vu, Roy Harrison, Di Liu, and Bill Bloss from the University of Birmingham; Alastair Lewis, William Dixon, Marvin Shaw, and Stefan Swift from the University of York. Siyao Yue, Liangfang Wei, Hong Ren, Qiaorong Xie, Wanyu Zhao, Linjie Li, Ping Li, Shengjie Hou, and Qingqing Wang from IAP; Kebin He and Xiaoting Chen from Tsinghua University, and James Allan from the University of Manchester for providing logistic and scientific support for the field campaigns.

**Financial Support**

This work was supported by the Leeds-York-Hull Natural Environment Research Council (NERC) Doctoral Training Partnership (DTP) Panorama under grant NE/S007458/1. The APHH Beijing campaign was supported by the Natural Environment Research Council (NERC) under grant NE/N006917/1.

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

Figure 1. OH-initiated and NO₃-initiated formation of IHN. The formation of 1,4-IHN is shown here, other IHN isomers, as well as additional reaction products, will also be formed.

Figure 2. NO₃-initiated formation of ICN. The formation of 1,4-ICN is shown here, other ICN isomers, as well as additional reaction products, will also be formed.

Figure 3. NO₃-initiated formation of IPN. The formation of 1,4-IPN is shown here, other IPN isomers, as well as additional reaction products, will also be formed.


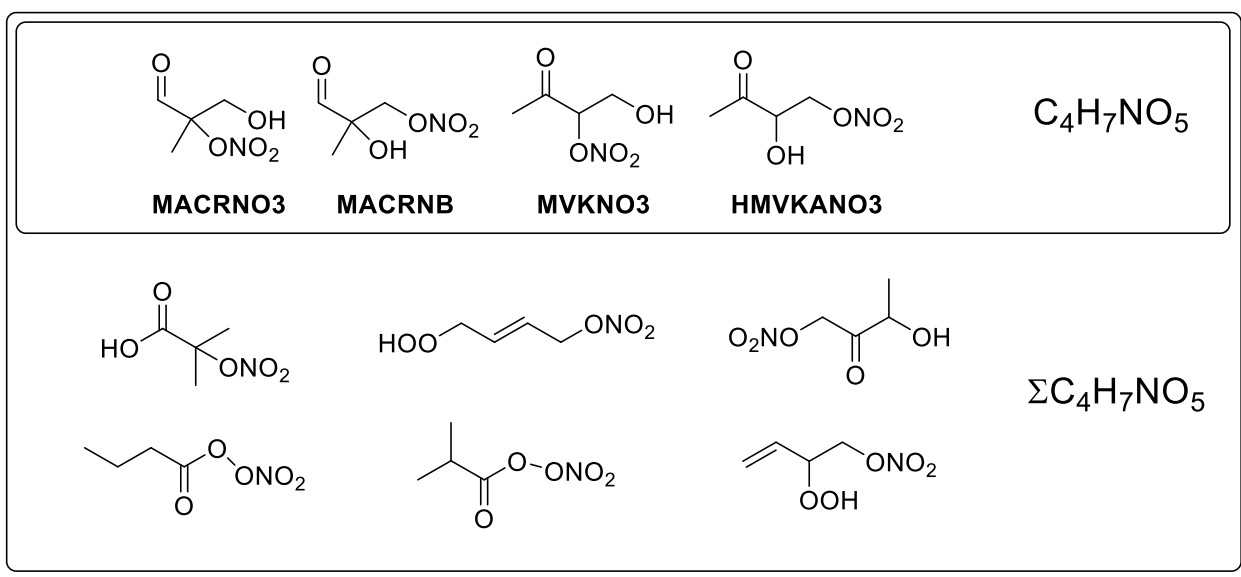


**Figure 4. The four C₄H₇NO₅ species resulting from isoprene oxidation present in the MCM along with the additional isomeric compounds which complete the set of ΣC₄H₇NO₅**

**Figure 5. Formation of C₄H₇NO₅ compounds. Only two isomers are shown here, other formation routes for these and other isomers are also present. Additional reaction products will also be formed.**


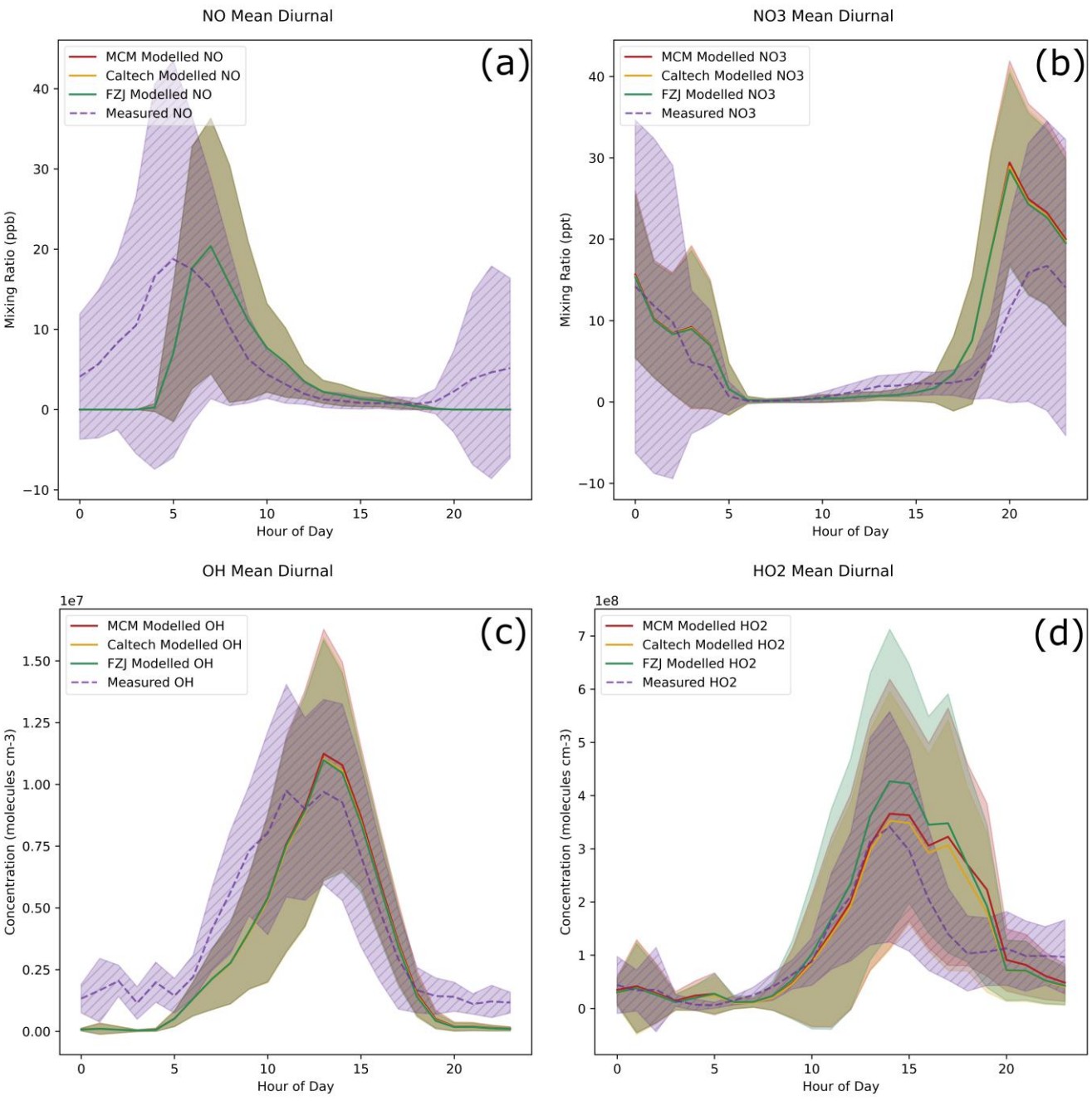

**Figure 6. A selection measured values and model predictions of inorganic species left unconstrained in the models. Each line shows the mean value for each dataset, with the shaded area indicating one standard deviation above and below the mean. The values of NO from each model are all overlapping in (a).**

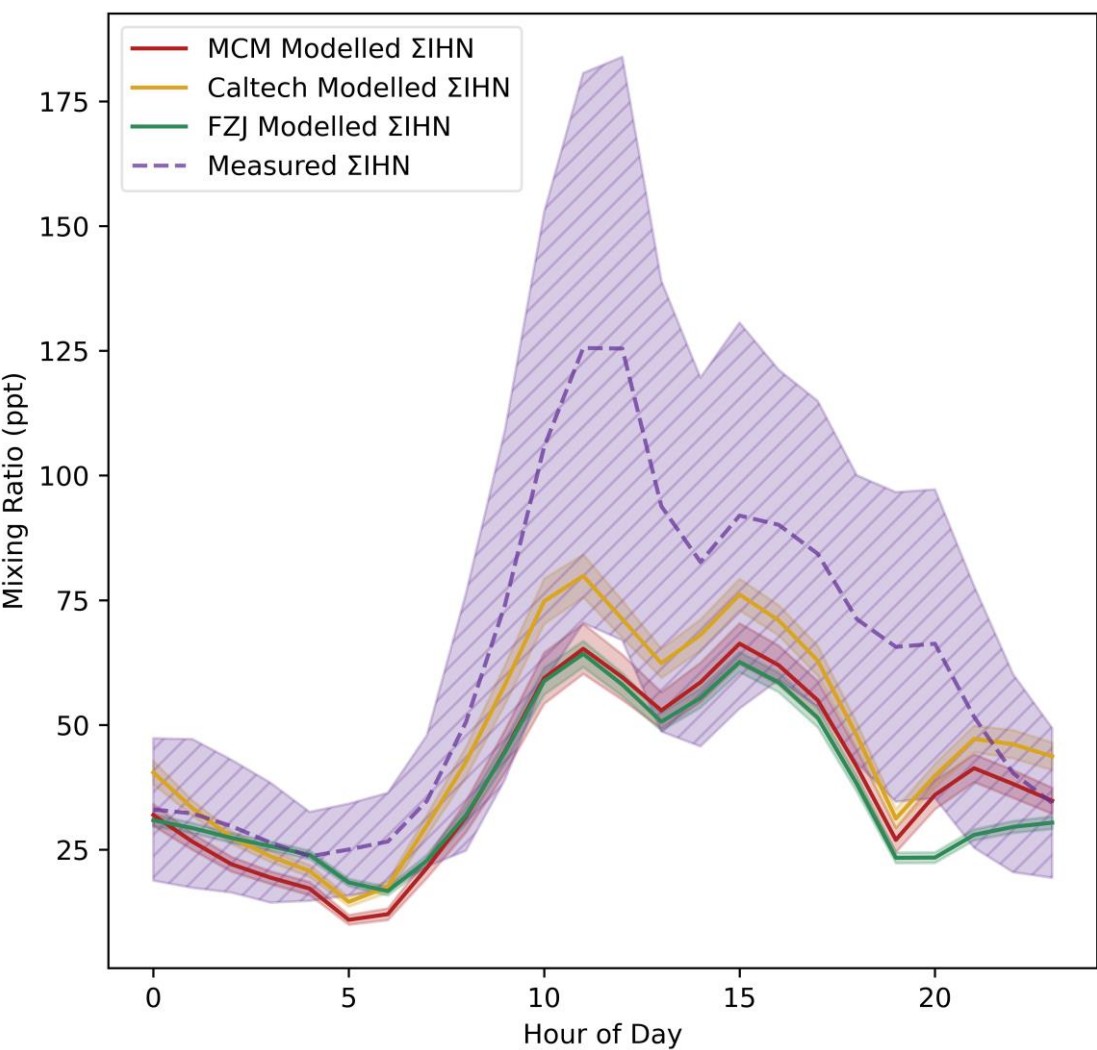

**Figure 7. Measured and modelled ΣIHN. Each line shows the mean value for each dataset, with the shaded area indicating one standard deviation above and below the mean.**


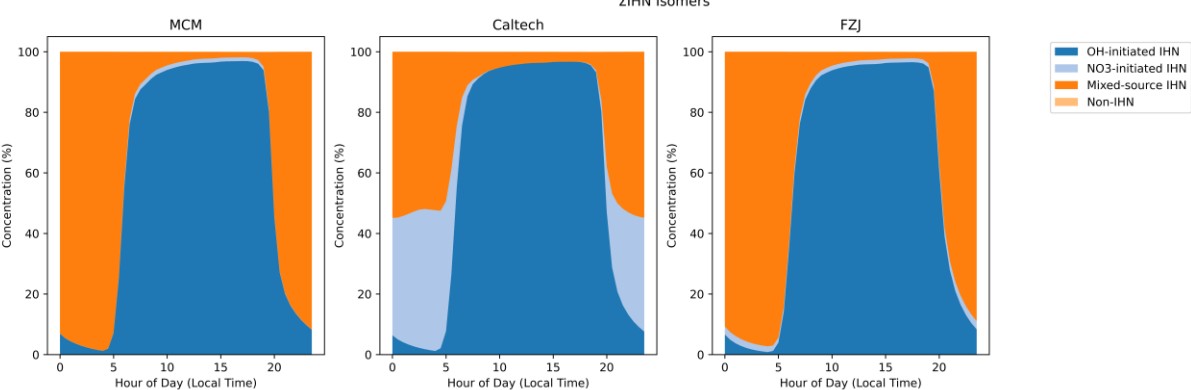

**Figure 8. Isomer composition of the modelled ΣIHN. OH-initiated IHN are those primarily formed by OH chemistry, the 1,2-IHN and 4,3-IHN. NO3-initiated IHN are those primarily formed by NO3 chemistry, the 2,1-IHN and 3,4-IHN. Mixed-source IHN is formed in large amounts by both routes, the E/Z-1,4-IHN and E/Z-4,1-IHN.**

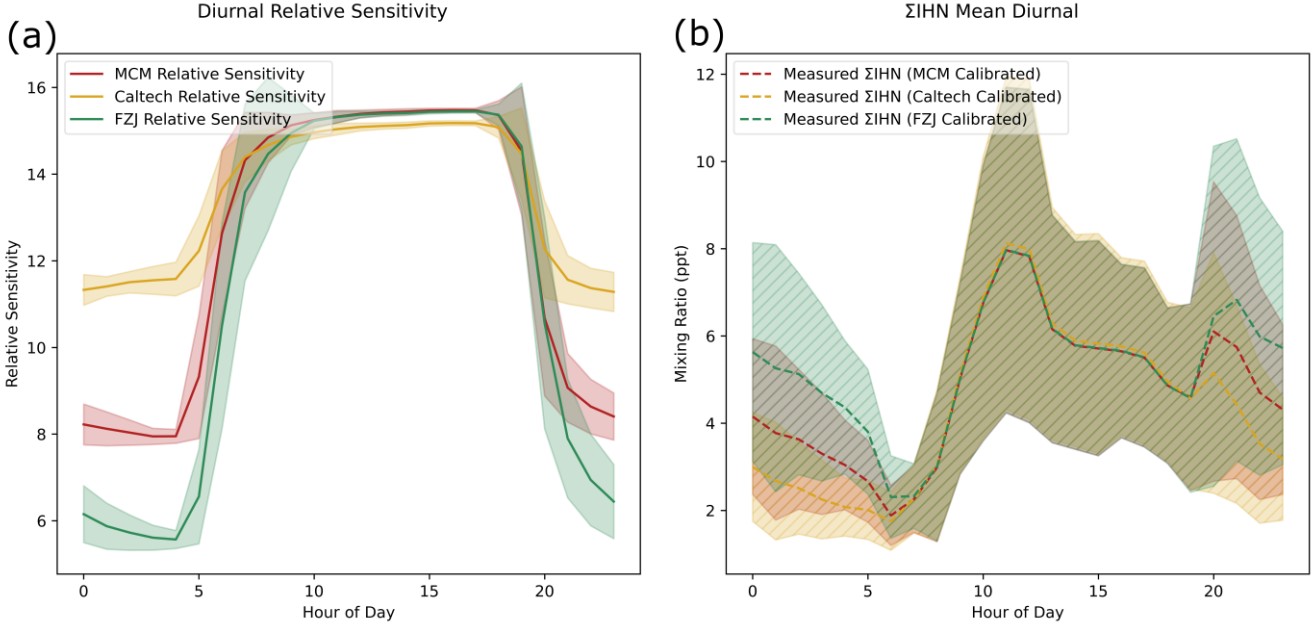

**Figure 9. (a) Diurnal variation in the sensitivity of I⁻-CIMS to ΣIHN relative to IEPOX according to the isomer distribution predicted by each model. (b) The measured ΣIHN data adjusted using the relative sensitivity values from each mechanism.**

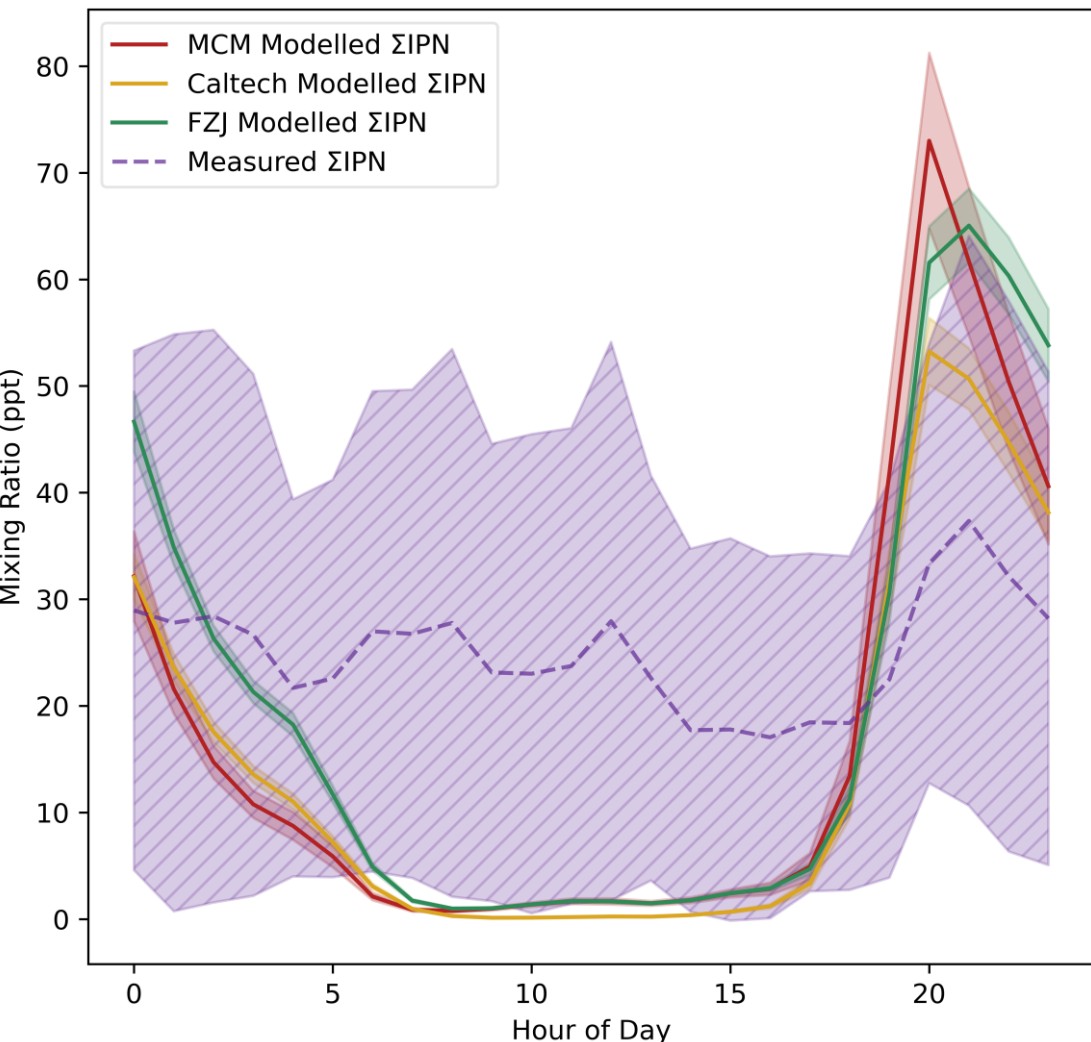


**Figure 10. Measured and modelled ΣIPN (a). Each line shows the mean value for each dataset, with the shaded area indicating one standard deviation above and below the mean.**

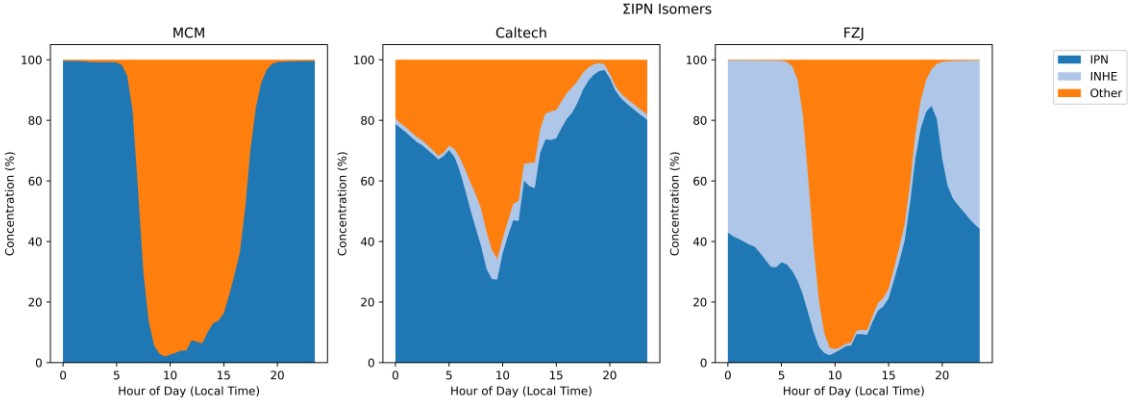

**Figure 11. Isomer composition of the modelled ΣIPN as a percentage of total ΣIPN. "Other" comprises of ISOP1N253OH4OH, C530NO3, PPEN, C524NO3, C51NO3, and C5PAN4.**

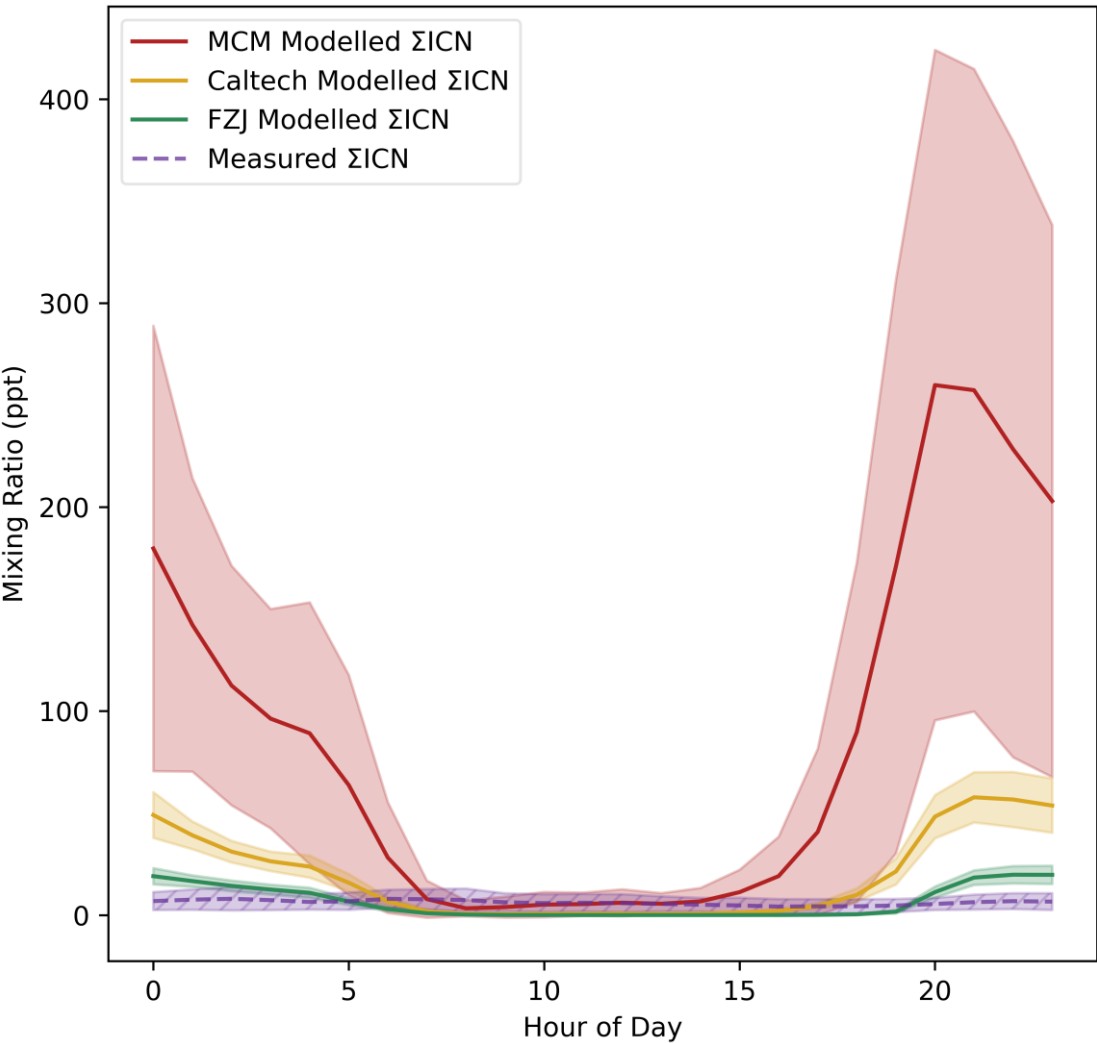

**Figure 12. Measured and modelled ΣICN. Each line shows the mean value for each dataset, with the shaded area indicating one standard deviation above and below the mean.**

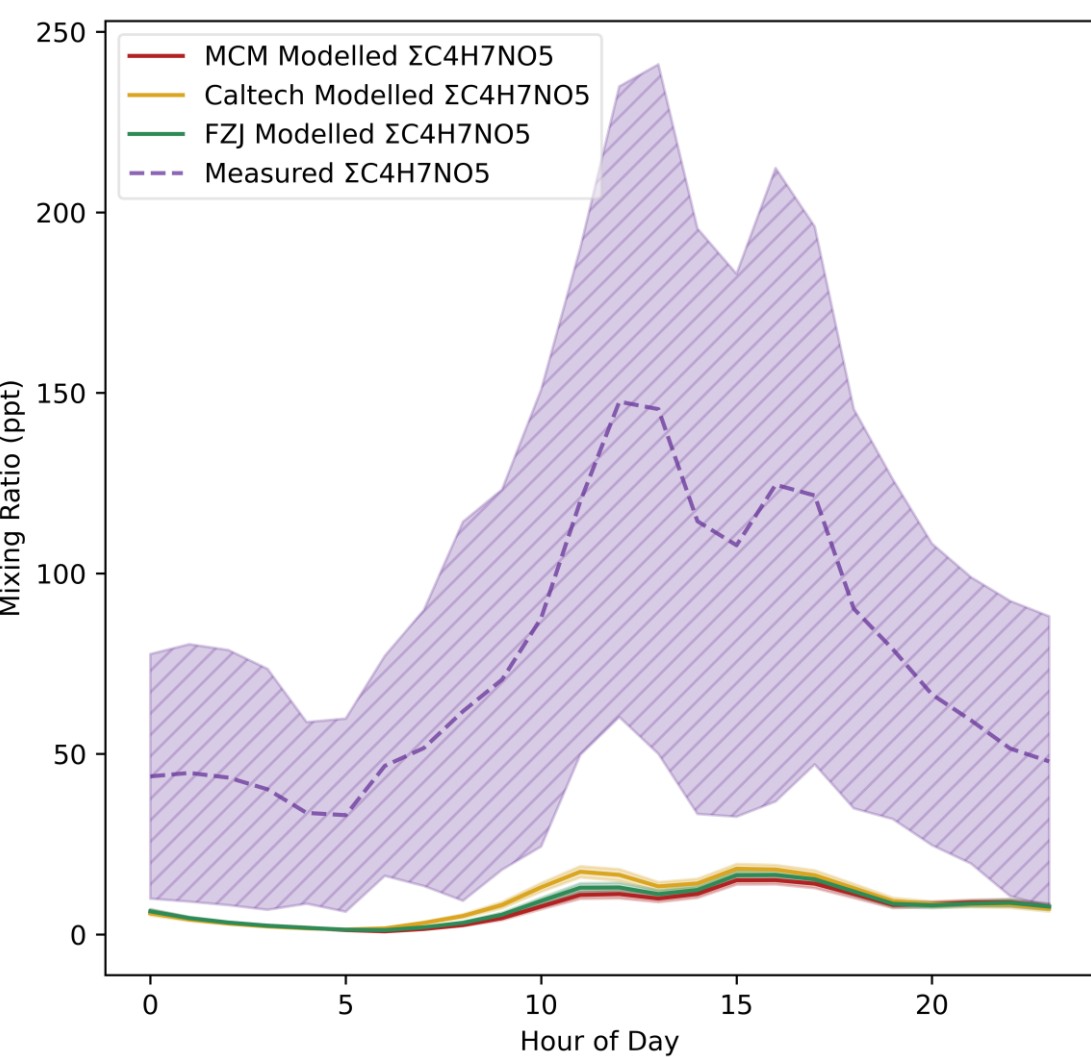


**Figure 13. Measured and modelled ΣC₄H₇NO₅. Each line shows the mean value for each dataset, with the shaded area indicating one standard deviation above and below the mean.**