# Peer review of "Evaluation of Isoprene Nitrate Chemistry in Detailed Chemical Mechanisms"

_Atmospheric Chemistry and Physics, 2022_

## Author Response (AR1)

Firstly, we would like to thank the reviewers for their comments. Both reviewers provided extremely valuable insight that has helped to vastly improve the quality and findings of the paper. We believe that the increased discussion of calibration has helped to improve the quality of the discussion, but also provided useful data for researchers looking to apply I⁻-CIMS data to isoprene measurements in the future. Also, the inclusion of improved ventilation in the models increases consistency with previous work. Some of the comments requiring large changes are collected here for ease of reviewing, these changes will then be referenced in the point-by-point response using the numbering system.

**1) Changing of Model Ventilation Term**

Both reviewers mentioned the overprediction of primary isoprene oxidation products and the differing representations of physical processes between this work and previous modelling work of the same campaign by Whalley *et al.* and Reeves *et al.* As pointed out, our 24-hr constant mixing lifetime is considerably longer than the variable mixing lifetimes used in the previous work. While we may not expect to require the exact same mixing rates, because we have different accounting of deposition processes, we have updated the models to use a diurnally varying mixing rate. In line with Whalley *et al*. and Reeves *et al.* this variable mixing lifetime is tuned to fit glyoxal concentrations as closely to measured values as possible. Changing the mixing in this way does alter the concentrations of our nitrates of interest, however we believe our conclusions are reasonably robust to these changes and the manuscript has been adjusted to reflect these changes where required. The major changes are in MVK/MACR and C4H7NO5, the modelled concentrations of both of these groups of compounds decrease and are under-predicted by the new models.

As a result of the changing ventilation, all of the paper's figures have been updated and the updated figures are added to the end of this document for clarity. The figure numbers of these figures has also changed and have been updated throughout the manuscript.

1a) We believe the MVK/MACR under prediction is explainable by its long lifetime, though this under-prediction may result from slightly too rapid mixing rates. This is explained in the relevant part of the Model Validation section which now reads as follows:

> "When comparing the modelled and measured MVK and MACR mixing ratios, while day-time concentrations are at-most half of the measured values, the night-time concentrations fall far below the measurements (Figure S2). This may be the result of the long lifetime of MVK and MACR, meaning there is a high background concentration not captured by the models. Alternatively, it may due to imperfect accounting for physical processes such as mixing and ventilation within the models or a poor understanding of MVK+MACR chemistry in this environment. There may also be some role played by the conversion of isoprene hydroxyhydoperoxides to MVK+MACR on the metal inlets of the mass spectrometers resulting in an artificially increased measurement. (Rivera-Rios et al., 2014; Newland et al., 2021) It is also important to consider the effect of upwind isoprene concentrations for all of the isoprene oxidation products discussed in this work. While our modelling makes use of isoprene concentrations measured at the same site as the product measurements, the upwind isoprene concentrations would be more useful for predicting the concentrations of isoprene oxidation products."

1b) The C4H7NO5 under prediction is stark, though some under prediction may result from the MVK+MACR under prediction, as well as this species' own long lifetime. The C4H7NO5 section now reads as follows:

"$\Sigma C_4H_7NO_5$ mixing ratios are under-predicted by around an order of magnitude in all models (Figure 13). The modelled $\Sigma C_4H_7NO_5$ diurnals only slightly vary between each model, despite the additional dark formation rates added to the FZJ mechanism, with the Caltech mechanism actually producing the highest concentrations. This is because the formation of $\Sigma C_4H_7NO_5$ is dominated by the OH oxidation of MVK and MACR. The time series for measured and modelled $\Sigma C_4H_7NO_5$ is given in Figure S15.

The under-prediction in MVK+MACR and the potentially high ventilation (see Section 3.1) may account for some of this under-prediction, particularly in light of the potentially long lifetime of $C_4H_7NO_5$, however the under-prediction is much stronger than is observed for the MVK+MACR precursors. (Müller et al., 2014) Without previous work investigating the sensitivity of I⁻-CIMS to $C_4H_7NO_5$ it is difficult to assess the impact of calibration on this measurement. Assuming a similar sensitivity as the most sensitively detected IHN isomer, where the OH and $NO_3$ groups are in close proximity like in the $C_4H_7NO_5$ isomers, would bring the measurement in line with the models."

1c) The description of the ventilation term in section 2.3 now reads:

"Additionally, a loss term was included for all species to account for mixing and ventilation. A diurnally varying ventilation rate was applied, where the rate was scaled such that the modelled glyoxal concentrations matched measurements, in a similar fashion to previous work. (Whalley et al., 2021; Reeves et al., 2021) The sensitivity of the model results to this term is assessed in the Model Validation section."

1d) The description of HO2 in the Model Validation section now reads:

"Day-time $HO_2$ concentrations are around 2 times higher than the measurement during the evening in all models (Figure 6d), which is…"

1e) The description of the ventilation sensitivity test statistics has been updated and also changed from % to fractional changes to be consistent with the fractional changes discussed throughout the rest of the paper. The sentences now read:

"The halving of the ventilation rates resulted in an average change in concentration across the models run with each mechanism of 3.1, 1.5, 1.8, and 1.8 times for $\Sigma C_4H_7NO_5$, $\Sigma$IHN, $\Sigma$ICN, and $\Sigma$IPN respectively. The average changes for doubling the ventilation rate were 0.32, 0.62, 0.60, and 0.56 for $\Sigma C_4H_7NO_5$, $\Sigma$IHN, $\Sigma$ICN, and $\Sigma$IPN respectively."

1f) The IEPOX+ISOPOOH description in the model validation section has been updated to reflect the underprediction in IEPOX+ISOPOOH seen. It now reads:

[revised manuscript text omitted]

2b) A discussion of IPN calibrations has been added to the end of the IPN section that reads:

"As with all of the nitrates investigated here, the role of the I⁻-CIMS calibration on the data presented must be considered. As shown previously, all models predict a diurnally varying isomer distribution with night-time ΣIPN being largely comprised of IPN and/or INHE, and daytime ΣIPN being comprised of smaller concentrations of other species. If the daytime isomers were much more sensitively detected than the night-time isomers then this could offset the diurnal concentration profile modelled to produce a constant measured signal throughout the day, as is observed. The daytime ΣIPN concentrations predicted by the MCM and FZJ models is around 0.06 times the measured values, meaning that the daytime isomers would need to be around 17 times more sensitively detected than IEPOX to reproduce the flat diurnal signal observed, assuming the night-time isomers had the same sensitivity as IEPOX. There has been very little research to quantify the sensitivity of I⁻-CIMS to hydroperoxides, but Lee *et al.* reported the sensitivity of peroxyacetic acid to be 0.04 times that of acetic acid suggesting that the non-hydroperoxide daytime nitrates may be more sensitively detected than the night-time IPN. (Lee et al., 2014)"

2c) A discussion of ICN calibration has been added to the end of the ICN section that reads:

"While this account of increasingly complex alkoxy radical chemistry gives good reason to question the high ICN formation rates from the MCM, it is also important to consider that previous work has found the lower sensitivity to aldehyde and ketone groups by I⁻-CIMS compared to alcohols, as such it should be expected that the measured ΣICN is most likely to be under-quantified by use of the IEPOX calibrant compared to species such as IHN.(Lopez-Hilfiker et al., 2014; Iyer et al., 2016; Lee et al., 2014) For example, Lee *et al.* 2014 shows that the sensitivity to hydroxyacetone is around 20 times lower than the similarly structured 1,2-butanediol and the sensitivity to 2,5-hexanedione is around 70 times lower than that of 5-hydroxy-2-pentanone. Assuming the relative sensitivity of ICN to IEPOX is lower than that of IHN, i.e. the sensitivity relative to IEPOX is lower than 15.64 (Section 3.2.1), would mean that the over-prediction made by the MCM could not be solely accounted for by the calibration. However, it is more difficult to comment on the accuracy of the FZJ mechanism compared to the Caltech mechanism in this respect as a reasonable calibration correction could bring the measurement in line with either model."

2d) The note about authentic standards at the start of the conclusions has been changed to read:

"…unaccounted losses for species such as INHE. While the impact of I⁻-CIMS sensitivity on measurements of these nitrates has been considered throughout this work, the availability of authentic standards would greatly improve the ability to quantify such organonitrates."

2e) The description of reactive uptake on INHE in the conclusions has been changed to:

[revised manuscript text omitted]

**3) IHN Hydrolysis**

Finally, both reviewers mentioned the potential for IHN hydrolysis to account for elevated IHN concentrations. These comments have been addressed by adding a test of the models' sensitivity to hydrolysis of 1,2-IHN in a similar fashion as was performed for INHE.

3a) The following text was added to the IHN subsection:

"Previous work has shown that the hydrolysis of 1,2-IHN occurs rapidly in the atmosphere. (Vasquez *et al*., 2020; Liu *et al*., 2012) To test the sensitivity of our results to 1,2-IHN hydrolysis, loss reactions of 1,2-IHN were added to each of the mechanisms with a rate calculated as described in Section 2.3.1. Figure S6 shows the modelled ΣIHN using each of the mechanisms with 1,2-hydrolysis reactions included. Since the majority of daytime ΣIHN is comprised of 1,2-IHN, removal of this compound can have a large effect on the modelled ΣIHN. A $γ_{IHN}$ value of 1 removes most, but not all, of the 1,2-IHN and a value of 0.1 brings modelled ΣIHN concentrations close to when the value is 1. Conversely, $γ_{IHN}$ values below 0.01 only result in small changes to modelled ΣIHN compared to the base model where no IHN hydrolysis is included."

3b) The description of the particle-uptake parameterisation has been moved to the experimental section in a new subsection (Section 2.3.1). This section reads as follows and allows for a more succinct discussion in the IHN and IPN sections:

"In the cases of ΣIHN and ΣIPN, an analysis of the impact of the particle-phase hydrolysis of 1,2-IHN and the reactive uptake of INHE is performed. For both of these cases, the rates of loss ($k_{IHN}$ and $k_{IHNE}$ for IHN hydrolysis and INHE uptake respectively) are calculated using Equation 1. $S_a$ is the aerosol surface area, as calculated for each model time-step from scanning mobility particle sizer (SMPS) measurements, $r_p$ is the effective particle radius calculated as a weighted median of the SMPS number measurements at each model time-step, $D_g$ is the gas-phase diffusion coefficient, $v$ is the mean molecular speed of IHN or INHE molecules in the gas phase, and $γ$ is the reactive uptake coefficient. $v$ was calculated using Equation 2 where R is the ideal gas constant ($8.314$ $J$ $K^{-1}$ $mol^{-1}$), T is the measured

temperature at each time-step, and $M_r$ is the molecular mass of the compound of interest (0.147 kg mol$^{-1}$ for IHN and 0.163 kg mol$^{-1}$ for INHE). A value of $1\times10^{-5}$ m$^2$ s$^{-1}$ was used for $D_g$, as is assumed in Gaston *et al.* for IEPOX. (Gaston et al., 2014) This method has been extensively used to calculate the rate of reactive uptake of IEPOX. (Gaston et al., 2014; Riedel et al., 2016; Budisulistiorini et al., 2017)

$$k_{IHN} = \frac{S_a}{\frac{r_p}{D_g} + \frac{4}{v\,\gamma_{IHN}}} \qquad\qquad Equation\ 1$$

$$v = \sqrt{\frac{3\,R\,T}{M_r}} \qquad\qquad Equation\ 2$$

An estimation of γ is complicated by the dependence on particle properties. In each case, results are shown for models where a range of γ values are assumed, between the limits of 0 and 1."

3c) Since Equation 1 has been moved to the Experimental Section, the description of particle uptake in the IPN section now reads:

"In order to assess the potential for reactive uptake of INHE on the modelled ΣIPN, loss reactions for each of the four INHE isomers in the FZJ Mechanism were added to the mechanism and the models rerun. The rate coefficient for the reactive uptake of INHE ($k_{INHE}$) was calculated as described in Section 2.3.1. Figure S11 shows the modelled ΣIPN produced by a set of models for which a range of $\gamma_{INHE}$ were assumed, between the limits of 0 and 1. When $\gamma_{INHE}$=1 and $\gamma_{INHE}$=0.1, almost all…"

**Reviewer 1**

This manuscript describes an intercomparison between measured isoprene-derived organonitrate species from a polluted megacity in China and simulated mixing ratios of the same organonitrates from box models with three detailed isoprene chemistry mechanisms. Because isoprene is such a critical volatile organic compound (even in urban areas), and because the removal of reactive nitrogen species via organonitrate formation from VOCs like isoprene can play a crucial role in regulating ozone formation, oxidizing capacity, and particulate formation, the accurate modeling of these processes is highly important for simulations of air quality.

The manuscript is quite clear and well-written, and effectively guides the reader through the process and outcomes of the research topic. Some surprisingly large differences arise between the three state-of-the-art mechanisms, but they are clearly described and their impacts well-enumerated. The sensitivity analysis of the INHE uptake term is particularly compelling. However, some aspects of the model-measurement comparisons remain unconvincing, and in particular, how much the reader should read into certain model-measurement discrepancies isn't clear. The manuscript lacks a quantitative assessment of measurement uncertainties even though that very uncertainty -- or, at least, the potential for instrumental sensitivity to the compounds of interest to vary over time due to varying contributions of isobaric isomers -- becomes a crucial message of the manuscript (and one that I think deserves mention in the abstract). More of the manuscript is devoted to the potential for various model processes to influence results, such as ventilation timescales and INHE uptake, but two factors that seem of critical importance for determining model outcomes -- namely, the aqueous hydrolysis of tertiary nitrates and the potential for model-measurement differences in HO2 and NO to affect RO2 fates -- are not quantitatively discussed, which limits the applicability of these results beyond the confines of the present box-model analysis. More detailed questions on these issues are included in the line-numbered comments below.

Finally, it would be very interesting to know what the models determine the fate of the analyzed organonitrates to be, considering that this determines their major impacts on air quality and atmospheric chemistry. To what extent is NOx recycled back to the gas phase or transported out by ventilation? While this could of course open another proverbial can of uncertainty worms, it might at least be worth a mention, especially if there are differences between the mechanisms or between the species analyzed (i.e. IPNs vs. IHNs vs. ICNs).

> The losses of these nitrates in the models are largely dominated by physical processes, at over 40% of the total losses at all times, and normally over 80% at night-time. However, the chemical losses are often not well described in the mechanisms, often only including loss by reaction with OH, and the Caltech and FZJ mechanisms often forming species without any chemical losses. The MCM tends to offer more speculation as to the reaction of these compounds with $NO_3$ and $O_3$ based on SAR predictions and will always offer some (photo)chemical loss routes for organic species. An analysis of the role of additional chemical losses is provided for IPN, in response to Reviewer 1's comment on L216-217. Additionally, a comment has been added to the conclusions section noting the uncertainty surrounding the losses of these nitrate species:

"While physical processes dominated the loss of the organonitrates in all of the models presented here, the chemical losses of these species are not well understood. Estimated rate constants for the reaction of IHN, IPN, and ICN from Wennberg *et al.* indicate that the OH reactions which are included in all of the mechanisms may be the major chemical loss pathways, with $NO_3$ oxidation comprising a larger loss than reaction with $O_3$. This has implications for $NO_x$ recycling, indicating that most of the $NO_x$ consumed to form the organonitrates is subsequently lost from the gas-phase or transported away from the site of formation. (Bates and Jacob, 2019)"

L 109-110: Can some discussion be provided here about how much uncertainty is introduced by using a single invariant calibration factor for all organonitrate species in the I- CIMS and, on top of that, one that is derived from a non-nitrate compound? In general it would be helpful throughout to add more discussion of the measurement uncertainties when comparing with the models, so that readers can be aware of instances when the model-measurement disagreement may not be statistically significant. It would also be immensely helpful to show the measurement uncertainty on some of the figures, although I understand this would be difficult to combine with the bounds already shown to represent the standard deviations across days.

It is extremely difficult to assign measurement uncertainties to the I-CIMS in light of the lack of authentic standards. The additional discussion of calibrations included throughout the manuscript (2a-j) (particularly in the added IHN section, 2a) gives the reader a better understanding of the uncertainty in the I-CIMS data. We believe it is best to keep the diurnal plots with shaded areas representing the day-to-day variability rather than an uncertainty value as any uncertainty bounds estimated would be very tentative whereas the standard deviation of the diurnal mean is a more specific parameter.

L 201-208: The daytime ISOPOOH+IEPOX overestimate is likely attributable in part to the model overestimates of HO2, which therefore emphasizes the RO2+HO2 pathway more than measurements suggests. (However, NO is also overestimated in the afternoon, it appears, so I can't be sure of the balance of these compensating errors). It would be interesting to note here that this also suggests the RO2+NO pathway may be underestimated in the models, which would exacerbate daytime overestimates of IHNs. This leads to two points that I think deserve more discussion:

- first, it looks like the sum of *all* major isoprene first-generation products are drastically overestimated in the afternoon, when MVK, MACR, IEPOX, ISOPOOH, and the nitrates are combined. Could this just be a result of excess isoprene in the model? I see that model isoprene is constrained to the measurements, but perhaps it is the upwind isoprene, not the in situ isoprene, that matters more here.

While it is true that the upwind isoprene will be a more important factor in the concentrations of these nitrates, we have no measurement of the upwind isoprene concentrations and so cannot include this in the modelling. In any case, the revised ventilation scheme (1a-k) eliminates this over-prediction for the majority of compounds. A note has been added to the Model Validation section to acknowledge this potential source of error (1a).

- second, I think the reasoning behind not constraining NO and HO2 to measurements is well-described and sound, but it would be worth at least mentioning how different the product distribution would be if these crucial determinants of RO2 pathway were modeled correctly or

constrained to measurements. How much of the afternoon model overestimate of ISOPOOH or of IHN can be explained by the model overestimates of HO2 and NO respectively?

Models were run constrained to HO2 but the impact on the conclusions was minimal, especially since the new updates to ventilation improved HO2 predictions, meaning this was omitted from the paper. The nitrate diurnals from the HO2-constrained runs are shown below.

[Figure]

The model run constrained to NO produced more different diurnal profiles for the nitrates, but also worsened the predictions of NO3. This is why we originally chose to leave NO unconstrained, as mentioned in the manuscript. Showing the good model-measurement agreement for NO measured at 100m demonstrates that the model is reproducing NO concentrations away from the influence of local sources. The NO3 plot from the NO-constrained run is reproduced below.

[Figure]

We have chosen to omit the description of these runs from the manuscript for clarity, but will add them if the reviewer deems it necessary.

L 213: Terminal losses of tertiary nitrates to aqueous particles can be very rapid (Vasquez et al, 2020), to the extent that under humid, particle-rich conditions this can be the dominant IHN loss pathway. (The effect on other nitrates, like IPNs and ICNs, is not as well characterized, but could still be significant). It seems that this could be incorporated into the box models here with a similar (or even simpler) method to the INHE uptake parameterization, but even if the goal is to avoid doing more simulations, the potential contribution of this pathway should at least be estimated. To what extent could this hydrolysis correct the overestimate in IHN? If other functionalized tertiary nitrates behave similarly, how might hydrolysis affect the modeled ICN, IPN, and C4H7NO5 mixing ratios? And finally, given that the hydrolysis rates seem so isomer-dependent, how well is an isomer-lumping mechanism (like MCM for the IPNs) able to properly simulate this process?

As mentioned in the introduction, this comment (along with similar comments from Reviewer 2) has led us to add a sensitivity test of IHN hydrolysis to the paper (3a). Hydrolysis of the other compounds was not investigated due to the lack of previous work exploring their hydrolysis.

L 216-217: The IPN isomers (excluding isobaric C51NO3, INHE and dihydroxy-nitrooxy-isoprene) have double bonds, which means they should react with ozone and NO3 fairly rapidly. Is this really not included in any of the models? From the mean nighttime levels of NO3 and O3, can the contribution of these potential losses be estimated?

It is true that $O_3$ and $NO_3$ oxidation of IPN is not included in any of the mechanisms. The Wennberg 2018 review does provide estimated rates of reaction for IPN (alongside IHN and ICN) with $NO_3$ and $O_3$, but these rates are not included in the mechanism. These listed rate constants have been used alongside the measured OH, $O_3$, and $NO_3$ concentrations to estimate the average relative contributions of each oxidant to night-time chemical loss. This is explained in text added to the IPN section:

"While none of the mechanisms include $NO_3$ or $O_3$ oxidation of IPN, the Wennberg *et al.* 2018 review of isoprene chemistry does list estimated reaction rates of IPN, ICN, and IHN with $NO_3$, $O_3$, and OH. (Wennberg et al., 2018) Figure S8 shows the average proportional night-time chemical loss for IHN, IPN, and ICN calculated using the rates given in Wennberg *et al.* and the measured OH, $O_3$, and $NO_3$ concentrations between 20:00 and 05:00. For the IPN isomers, OH oxidation accounts for the majority of the chemical loss of IPN at night, with around 10-15% being lost to reaction with $NO_3$. Reaction with $O_3$ also makes up a substantial fraction of the chemical loss in the 1,4-IPN and 4,1-IPN isomers, though OH is still the major chemical sink. Since OH oxidation is included in the mechanisms, then the majority of the chemical losses should be captured by the models. Physical processes also dominate the losses of ΣIPN at night, so the addition of more chemical losses would not have a large impact on ΣIPN concentrations."

L 217: The reason given here for the modeled IPN diurnal profile is the lack of nighttime loss processes, but that would have the opposite effect from what the models show, which is a gradual but substantial decrease over the course of the night (after the sunset spike) resulting in a minimum at sunrise. If there are no nighttime loss processes, is this gradual decrease due entirely to the mixing-out lifetime, and why is the rapid loss relatively insensitive to the mixing out rates (Fig S3)? It seems, both here and for IHN in figure 9, that the modeled nighttime loss rates are too high (although this may, of course, be alternatively attributed to nighttime sources being too low) -- how can they be reduced?

The comment regarding the lack of night-time IPN losses was meant as an explanation of the changing concentrations in the model, rather than an explanation of the model-measurement discrepancy. The strong diurnal results from the main formation being at night-time, when there is very little chemical loss. The line has been changed to reflect this:

"This is because the majority of IPN is formed through $NO_3$ oxidation of isoprene at night when there are few losses. The only losses of IPN in all mechanisms, besides the added deposition reactions, are photolysis reactions and the reaction with OH."

With the new ventilation scheme, ΣIPN concentrations are shifted down such that the night-time concentrations are close to the measured value, while day-time concentrations are under-predicted. This changes the discussion of the model-measurement comparison somewhat. We have provided a discussion of this comparison, largely through the discussion of calibration (2b). Since the losses are dominated by physical processes which was determined by literature comparison and comparison of glyoxal concentrations, we have simply noted the potential role of physical processes in the underprediction (1h).

L 295-308: This potential diurnal variation in calibration factors is very interesting and potentially important both for the conclusions of this paper and the wider community; I'd suggest including a reference to it in the abstract. The varying calibration factor was not applied to the I- CIMS measurements reported here, was it? Can any quantitative estimate be provided here for how much difference the application of a time-varying calibration factor would make to the measurements reported here and shown in Figure 9? Also, to what extent might the same issue of variable sensitivity come into play for the other compounds measured and reported here -- e.g., the fact that some species isobaric with IHNs (MPRKNO3, MIPKBNO3...) contain carbonyls rather than hydroxyl groups (reducing sensitivity, I believe), and the fact that some C4H7NO5 species are hydroxy-carbonyl-nitrates while others are PANs, nitrooxy-acids, or hydroxperoxy-nitrooxy-alkenes?

As noted in the introduction to this response, the discussion of calibration has been increased throughout the manuscript as result of this comment and similar points raised by Reviewer 2 (2a-j). We have included a reference to calibration in the abstract (2j). We have applied the varying calibration factor to the I-CIMS measurement of IHN in the new IHN subsection (2a), thus giving a quantitative estimate of the effect of the varying calibration factor. We have also noted the potential impact of calibration errors for the other nitrates where available (2b-c, 1b).

L 319-322: Is there any quantitative estimate of the sensitivity difference that can be provided here? Could it be a big enough difference to bring any of the models into agreement with measurements?

We have added a quantitative comparison to the IHN sensitivity (2c). We believe that the sensitivity differences between ICN and IEPOX may account for the FZJ or Caltech over-predictions, but should not be expected to account for the MCM over-prediction.

Figure 5: I don't think that ozonolysis in the top section is correct; ozonolysis should break the double bond, which would not result in any C4 fragments. (Ozonolysis of 3-hydroxy-4-nitrooxy isoprene would work here though). Also, why are there no co-reactants on the bottom pathway?

The co-reactants on the bottom pathway were omitted in error and have now been added. The ozonolysis reaction has also been adjusted to show the correct isomer.

Figure 6: Are the different modeled NO traces overlapping, or are some missing? If they're overlapping, that's probably worth mentioning in the caption just to avoid confusion.

The lines are overlapping, a note has been added to the figure caption explaining this.

Fig S10: The legend seems to say MVK+MACR where it should say ICN.

This has been corrected.

**Reviewer 2**

This paper nicely compares three different complex chemical mechanisms to explore how each represents organic nitrates from both OH and NO3 oxidation of isoprene. This study is quite useful and interesting to show the differences between these mechanisms. However, as explained below there needs to be more clarity in how dilution was constrained in the model and better calibration of the main isoprene organic nitrates including isoprene carbonyl nitrate (ICN), isoprene hydroxy nitrate (IHN), and isoprene hydroperoxy nitrate. Assuming that the sensitivity of all isomers and all organic nitrate types regardless of functional groups is similar to IEPOX very likely leads to inaccurate conclusions in the overall magnitude and even the diurnal pattern of these organic nitrates, which makes it difficult to use the measurements to assess, which mechanism is correct, which seems to be the purpose of this study. Major revisions to include a more complex calibration for these isoprene organic nitrates especially IHN and ICN, which have been previously calibrated by other I- CIMS, are needed as explained further in the specific comments below prior to publication.

Page 3 line 91 – There are a couple versions available from the code repository referred to in Wennberg 2018. Can you be clearer which version you used here? Both number and if it was full/reduced?

> "full v5" was added to the description of the Caltech isoprene mechanism, corresponding to the title and version number of the mechanism used.

Page 4 line 110 – Please further explain the sensitivity/calibration assumptions used here. What is the rationale to use IEPOX calibration for all types of organic nitrates (IHN, ICN, IPN) and all isomers? I recognize calibrations of IPN are uncertain as no standards are available, but IHN has been calibrated for several other I- CIMS instruments (Xiong et al., 2015 and Lee et al., 2014 (https://doi.org/10.1021/es500362a) and less, but still some information is available for ICN too also using an I- CIMS (Xiong et al., 2016, https://doi.org/10.5194/acp-16-5595-2016). These three papers demonstrate that different isomers and functional groups can cause very different sensitivities in the I- CIMS for these organic nitrates. Can you use the isomer distribution from the models and the isomer dependent sensitivities from these past works to more accurately calculate the measurements of these organic nitrates from the I- CIMS? Please provide either significant justification for not doing this with an estimate for uncertainty added or use a more complex assumption for the sensitivities of all the isoprene derived nitrates, but especially IHN, which has already been well studied by I- CIMS.

> While it is true that prior work has used authentic standards to analyse IHN and ICN by I- CIMS, in all cases this prior work has used specially synthesised standards which we did not have access to.

> As noted in the introduction to these responses (2a-j), we have taken the reviewer's suggestion of using the isomer distribution from the models and the isomer dependent sensitivities to try and address the calibration issues for IHN (2a). We are fortunate that Lee et al. 2014 provides a sensitivity value for IEPOX alongside the IHN isomers which allow for this analysis. This same data is not available for ICN, and so the same approach cannot be taken, though we have tried to improve our discussion of ICN calibration in the ICN subsection by making a comparison to the expected sensitivity relative to IHN (2c). Discussion of calibration for IPN and C4H7NO5 have also been added (2b, 1b). The conclusions and abstract have also been updated (2f-j).

Page 4 line 125 – Can you explain how you calculated these RO2 reaction rates further? Perhaps an example would help. When you say you use an average of all RO2 reactions do you also add in the reactions with acyl peroxy radicals that have faster reaction rates? Another more consistent approach is to do something similar to what MCM assumes, which is the geometric mean of the rate of the self-reaction of RO2 + RO2 and rate of CH3O2 + CH3O2?
http://chmlin9.leeds.ac.uk/MCM/categories/saunders-2003-4_6_5-gen-master.htt?rxnId=4270.

> This is a very valuable comment that we are extremely grateful for. Upon reflection it is unreasonable to assume that the reaction of the isoprene RO2s with any generic RO2 would proceed at the rate of the average rate of the isoprene RO2 isomers. While there are no acyl peroxy radical reactions to deal with from the Caltech mechanism, switching to an approach consistent with the MCM did remove the night-time peak in IHN observed in the Caltech models. This second peak was the consequence of the averaging approach chosen and resulted in drastically increased ISOP1N2OH formation rate from ISOP1N2OO cross-reactions (as is noted in the manuscript). Since the MCM averaging approach seems more logical and is more consistent with the other two mechanisms, the mechanism has been changed to use this approach. The mechanism description in section 2.2 has been adjusted to read:
>
> > "The Caltech Mechanism was integrated with the MCM subset for the additional VOCs by producing lumped $RO_2$ cross-reactions using the approach outlined in Jenkin *et al*.(Jenkin et al., 1997) For each $RO_2$ species where explicit reactions are given, the geometric mean of the self-reaction rate and the $CH_3O_2$ self-reaction rate was used. If a self-reaction was not specified, then the $CH_3O_2$ self-reaction rate was used. Branching ratios were then applied to the alcohol-forming, carbonyl-forming, and alkoxy-forming reactions according to Jenkin *et al.*"

Page 4 line 125 - Can you provide the reaction mechanism files (or a Table in the supplement with the changes) for at least the Caltech mechanism used here since you made updates beyond what is available publicly? This is important for data/code transparency. Providing the reaction mechanism files for all three mechanisms would be best.

> The mechanism files have been uploaded to the University of York's dataset archive system and has been assigned a doi now referenced at the end of Section 2.2:
>
> > "Each of the mechanisms used in this work have been made available online (doi.org/10.15124/500474f7-6e69-47db-baf7-36310451fd15)."

Page 5 line 153 – Can you further explain this sentence: "For multifunctional compounds, the largest deposition velocity was selected." Selected from where: Table S3 or from Nguyen et al., 2015?

> This sentence has been moved to after the introduction of Table S3 and re-phased in order to clarify the procedure. It now reads:
>
> > "For multifunctional compounds, the largest deposition velocity of each of the functional groups present in the compound was selected from Table S3."

Page 5 line 154 – do you mean divided by here: "The rate of deposition was determined by multiplying the assigned deposition velocity by the measured boundary layer height." as listed in the user guide: https://github.com/AtChem/AtChem2/blob/master/doc/AtChem2-Manual.pdf page 16.

> This is correct. The sentence has been changed to read:

> "The rate of deposition was determined by dividing the assigned deposition velocity by the measured boundary layer height."

Page 5 line 157 – Why did you choose this constant dilution rate? Do you have a reference for this? How does the dilution rate used in this paper compare with other box-modeling studies in the same region or similar regions? The papers you reference above (Reeves et al., 2021; Whalley et al., 2021) that also did box-modeling for APHH used a diurnally varying dilution rate dependent on glyoxal and the ventilation lifetime was a lot shorter than that used in this work. Considering that even MVK + MACR, which should be reasonably well represented chemically, are overpredicted maybe dilution should be stronger? How did you evaluate/constrain this?

> As noted in the introduction to these responses, the representation of ventilation in the models has been brought in line with these previous papers (1a-k). An updated description of the model ventilation is given in the experimental is given (1c) and the manuscript results updated throughout (1a-b, 1d-k).

Section 3.3: See above comment, especially for IHN when several studies have demonstrated that the different isomers have different sensitivities in the I- CIMS and we know from the modeling that the distribution of isomers will change diurnally (Figure 10 and last paragraph of Section 3.3), assuming the sensitivity of all isomers is similar to IEPOX likely leads to inaccurate conclusions in the overall magnitude and even the diurnal pattern, which makes it difficult to use the measurements to assess, which mechanism is correct. As suggested above, please update the measurements to consider these isomer dependent sensitivities. Also 1,2-IHN has been shown in Vasquez et al., 2020 to have rapid hydrolysis on aerosols. Have you considered this in your modeling? If not, how would not considering this impact your results?

> As noted in the introduction to these responses (2a-j), we have added a new IHN subsection to quantify and discuss the differing and changing IHN calibration (2a).

> Also noted in the introduction, we have added to the IHN section to explore the impact of IHN hydrolysis (3a).

Page 10 Line 321 – If you know that IEPOX is not likely a good calibrant to represent ICN because the I- CIMS is more sensitive to alcohols than aldehydes/ketones, can you choose a different calibrant based on these past literature studies (those referenced in this paragraph or Xiong et al., 2016) to better represent the ICN sensitivity? Without a better calibration for ICN or estimate of uncertainty, it is hard to determine which mechanism is best representing this chemistry.

> Unfortunately, the only calibration performed on the instrument was with IEPOX and as such, an alternative calibration is not possible beyond the approach taken in the new IHN subsection (2a). Additional discussion of calibration issues has been added to this section to give a general idea of the magnitude of the potential underprediction in ICN resulting from the IEPOX standard by comparison of the expected sensitivity difference between IHN and ICN (2c).

Page 11 line 353 – As mentioned above, it is not enough to state that you "potentially have significant issues with calibration factors". That's maybe okay for a compound like IPN, which have few standards and no past studies addressing the sensitivity on the I- CIMS, but you "certainly" not "potentially" have significant issues with calibration factors for IHN and ICN, for which other studies have reported absolute and relative sensitivities for the I- CIMS that could be used in this work.

We believe that this comment has been addressed through our increased focus on calibration throughout the manuscript (2a-j), particularly in the IHN subsection (2a) where previous data on IHN sensitivities is much more available. The sentence mentioned from the conclusion has been changed as noted in 2d.

Figure 5, For the NO3-initiated oxidation of hydroxycarbonyls in Figure 5, please add oxidants/reactants above the arrows for clarity and consistency with other plots.

The reactants have been added to Figure 5 where missing.

**Additional Changes**

A note about the noise on the IPN measurement has been added to section 3.3:

> "The time series for modelled and measured ΣIPN is shown in Figure S7. The data presented in Figure S7 show that there is substantial noise in the ΣIPN data, which may also mask diurnal trends and indicates that the ΣIPN concentrations are close to the instrument's detection limit for these compounds."…

The GCxGC data has been removed from this work after concerns were raised regarding the quality of the data. Most of the compounds measured were assumed to be of sufficiently low concentration to be unimportant, however the monoterpenes previously measured by GCxGC did influence the model results (largely by reacting with night-time $NO_3$). As such the monoterpene sum measurement by PTR and SIFT have been used to constrain representative monoterpenes as is now explained in the experimental section:

> "The sum of monoterpenes measured by PTR-MS and SIFT-MS was used to constrain alpha-pinene and limonene in the models, assuming each compound comprised 50% of the total monoterpenes."

Table S2 has also been adjusted to reflect these changes:

**Table S1. List of VOCs (and their names in the MCM) constrained to measured concentrations in the models. The "Measurement(s) Used" column indicates which instrument's measurements were used to constrain each species in model runs: proton transfer mass spectrometry (PTR), selected ion flow tube mass spectrometry (SIFT), and dual-channel gas chromatography with flame ionization detection (DC-GC).**

| Compound | MCM Name | Measurement(s) Used | Compound | MCM Name | Measurement(s) Used |
|---|---|---|---|---|---|
| isoprene | C5H8 | DC-GC, SIFT, PTR | ethane | C2H6 | DC-GC |
| monoterpenes | APINENE, LIMONENE | SIFT, PTR | propane | C3H8 | DC-GC, SIFT |
| ethene | C2H4 | DC-GC, SIFT | n-butane | NC4H10 | DC-GC |
| propene | C3H6 | DC-GC, SIFT, | i-butane | IC4H10 | DC-GC |
| trans-2-butene | TBUT2ENE | DC-GC | n-pentane | NC5H12 | DC-GC |
| 1-butene | BUT1ENE | DC-GC | i-pentane | IC5H12 | DC-GC |
| i-butene | MEPROPENE | DC-GC | n-hexane | NC6H14 | DC-GC |
| cis-2-butene | CBUT2ENE | DC-GC | n-heptane | NC7H16 | DC-GC |
| trans-2-pentene | TPENT2ENE | DC-GC | n-octane | NC8H18 | DC-GC |
| cis-2-pentene | CPENT2ENE | DC-GC | benzene | BENZENE | DC-GC, SIFT, PTR |
| 1,3-butadiene | C4H6 | DC-GC, SIFT | ethylbenzene | EBENZ | DC-GC, SIFT, PTR |
| acetylene | C2H2 | DC-GC, SIFT | propylbenzene | PBENZ | SIFT, PTR |
| methanol | CH3OH | DC-GC, SIFT | toluene | TOLUENE | DC-GC, SIFT, PTR |
| ethanol | C2H5OH | DC-GC, SIFT | o-xylene | OXYL | DC-GC |

The dates of the campaign in the experimental description have been changed to match the true campaign data used which is between 2017-06-01 and 2017-06-18. The year of 2021 was added in error. Meaning the opening line of Section 2.1 now reads:

> "The Beijing measurements used in this work were collected at ground level at the Tower Section of the Institute of Atmospheric Physics (IAP) in Beijing, China, between 2017-06-01 and 2017-06-18.(Shi et al., 2019)"

The order of the IPN and IHN sections has been switched to allow for a detailed discussion of calibration issues that can be referred back to throughout the manuscript.

The IEPOX isomer used to calibrate the CIMS data has been specified, as this may be important for the discussion around relative sensitivities compared to Lee et al. 2014, who use the same isomer as us. The relevant line in the experimental section has been changed to read:

> "Each nitrate was calibrated assuming the same sensitivity as trans-beta-IEPOX, though the potential role of calibration on the measured nitrate concentrations is discussed throughout this work.(Hamilton et al., 2021)"

The mechanism statistics in Table S1 which were incorrectly calculated for the Caltech and FZJ mechanisms have been corrected. The table is now as below:

| Property | MCM | Caltech Mechanism | FZJ Mechanism |
| --- | --- | --- | --- |
| Number of Reactions | 10371 | 10435 | 11046 |
| Number of Species | 3443 | 3589 | 3730 |
| Number of $INO_2$ Isomers | 1 | 4 | 8 |
| Number of IPN Isomers | 1 | 4 | 4 |
| Number of ΣIPN Isomers | 6 | 11 | 13 |
| Number of IHN Isomers | 5 | 8 | 8 |
| Number of ΣIHN Isomers | 9 | 12 | 12 |
| Number of ICN Isomers | 1 | 3 | 3 |
| Number of ΣICN Isomers | 1 | 3 | 3 |
| Number of $C_4H_7NO_5$ Isomers | 4 | 4 | 4 |
| Number of $ΣC_4H_7NO_5$ Isomers | 10 | 10 | 10 |

**Updated Figures**

**Figure 1. OH-initiated and NO₃-initiated formation of IHN. The formation of 1,4-IHN is shown here, other IHN isomers, as well as additional reaction products, will also be formed.**

**Figure 2. NO₃-initiated formation of ICN. The formation of 1,4-ICN is shown here, other ICN isomers, as well as additional reaction products, will also be formed.**

**Figure 3. NO₃-initiated formation of IPN. The formation of 1,4-IPN is shown here, other IPN isomers, as well as additional reaction products, will also be formed.**

[Figure]

**Figure 4. The four C₄H₇NO₅ species resulting from isoprene oxidation present in the MCM along with the additional isomeric compounds which complete the set of ΣC₄H₇NO₅**

[Figure]

**Figure 5. Formation of C₄H₇NO₅ compounds. Only two isomers are shown here, other formation routes for these and other isomers are also present. Additional reaction products will also be formed.**

[Figure]

**Figure 6. A selection measured values and model predictions of inorganic species left unconstrained in the models. Each line shows the mean value for each dataset, with the shaded area indicating one standard deviation above and below the mean. The values of NO from each model are all overlapping in (a).**

[Figure]

**Figure 7. Measured and modelled ΣIHN. Each line shows the mean value for each dataset, with the shaded area indicating one standard deviation above and below the mean.**

[Figure]

**Figure 8. Isomer composition of the modelled ΣIHN. OH-initiated IHN are those primarily formed by OH chemistry, the 1,2-IHN and 4,3-IHN. NO3-initiated IHN are those primarily formed by NO3 chemistry, the 2,1-IHN and 3,4-IHN. Mixed-source IHN is formed in large amounts by both routes, the E/Z-1,4-IHN and E/Z-4,1-IHN.**

[Figure]

**Figure 9. (a) Diurnal variation in the sensitivity of I⁻-CIMS to ΣIHN relative to IEPOX according to the isomer distribution predicted by each model. (b) The measured ΣIHN data adjusted using the relative sensitivity values from each mechanism.**

[Figure]

**Figure 10. Measured and modelled ΣIPN (a). Each line shows the mean value for each dataset, with the shaded area indicating one standard deviation above and below the mean.**

[Figure]

**Figure 11. Isomer composition of the modelled ΣIPN as a percentage of total ΣIPN. "Other" comprises of ISOP1N253OH4OH, C530NO3, PPEN, C524NO3, C51NO3, and C5PAN4.**

[Figure]

**Figure 12. Measured and modelled ΣICN. Each line shows the mean value for each dataset, with the shaded area indicating one standard deviation above and below the mean.**

[Figure]

**Figure 13. Measured and modelled $\Sigma C_4H_7NO_5$. Each line shows the mean value for each dataset, with the shaded area indicating one standard deviation above and below the mean.**

[Figure]

**Figure S1. Measured NO at 100m and modelled NO in each model. The mean values (a) show a peak before sunrise due to large spikes in the measurements in the morning on some days, so the median diurnal (b) is also shown.**

[Figure]

**Figure S2. Measured and modelled MVK+MACR mixing ratios. Each line shows the mean value for each dataset, with the shaded area indicating one standard deviation above and below the mean.**

[Figure]

**Figure S3. Impact on MVK+MACR (a), ΣC₄H₇NO₅ (b), ΣIHN (c), ΣICN (d), and ΣIPN (e) of varying the ventilation rate used in each model by 0.5 times and 2 times from the base mixing lifetime.**

[Figure]

**Figure S4. Measured and modelled ΣIEPOX+ISOPOOH mixing ratios. Each line shows the mean value for each dataset, with the shaded area indicating one standard deviation above and below the mean.**

[Figure]

**Figure S5. Time series for measured and modelled ΣIHN.**

[Figure]

**Figure S6. Measured and modelled ΣIHN mixing ratios for models using a range of γIHN values to account for the hydrolysis of 1,2-IHN. The mechanisms used in each model are as follows: (a) MCM, (b) Caltech Mechanism, (c) FZJ Mechanism. Each line shows the mean value for each dataset, with the shaded area indicating one standard deviation above and below the mean.**

[Figure]

**Figure S7. Time series for measured and modelled ΣIPN.**

[Figure]

**Figure S8. Proportional contribution of OH, O₃, and NO₃ to the night-time chemical loss (between 20:00 and 05:00) of IHN (a-h), IPN (i-l), and ICN (m-o) isomers. The loss rates are calculated using measured OH, O₃, and NO₃ concentrations and the rate constants listed in Wennberg et al. 2018.**

[Figure]

**C51NO3**          **C524NO3**          **ISOP1N23O4OH**          **ISOP1N253OH4OH**

**Figure S9. Structures of the three isomers of IPN that collectively comprise the majority of ΣIPN (C₅H₉NO₅) in the models.**

[Figure]

**Figure S10. Isomer composition of the modelled ΣIPN.**

[Figure]

**Figure S11.Measured and modelled ΣIPN mixing ratios for FZJ models using a range of γ_INHE values to account for the reactive uptake of INHE. Each line shows the mean value for each dataset, with the shaded area indicating one standard deviation above and below the mean.**

[Figure]

**Figure S12. Measured and modelled ICN relative to the concentration at 00:00.**

[Figure]

**Figure S13. Time series for measured and modelled ΣICN.**

**Figure S14. Examples of INO loss routes in each of the three mechanisms. Only one isomer is shown here, other isomers are present in the Caltech and FZJ Mechanisms. Additional reaction pathways are also possible in the Caltech and FZJ Mechanisms.**

[Figure]

Figure S15. Time series for measured and modelled $\Sigma C_4H_7NO_5$.